



# Statistical bias correction for climate change impact on the basin scale precipitation in Sri Lanka, Philippines, Japan and Tunisia

Cho Thanda Nyunt[1], Toshio Koike[2], and Akio Yamamoto[3]

[1]Department of Civil and Environmental Engineering, Hiroshima University, Hiroshima, Japan
[2]Department of Civil Engineering, The University of Tokyo, Tokyo, Japan
[3]Institute of Industrial Science, The University of Tokyo, Tokyo, Japan

*Correspondence to:* Cho Thanda Nyunt (nyunt11@hiroshima-u.ac.jp)

**Abstract.** We introduce a 3-step statistical bias correction method to solve global climate model (GCM) bias by determining regional variability through multi-model selection. This is a generalized method to assess imperfect GCMs that are unable to simulate distinct regional climate characteristics, either spatially or temporally. More remaining local bias is eliminated using an all-inclusive statistical bias correction addressing the major shortcomings of GCM precipitation. First, the multi-GCM
choice is determined according to spatial correlation (Scorr) and root mean square error (RMSE) of regional and mesoscale climate variation in a comparison with global references. After multi-GCM selection, there are three major steps in the proposed bias correction, i.e., the generalized Pareto distribution (GPD) for extreme rainfall bias correction, ranking order statistics for wet and dry day frequency errors, and a two-parameter gamma distribution for monthly normal rainfall bias correction. Best-fit GPD parameters are resolved by the RMSE, a Hill plot, and mean excess function. The capability of the method is examined by
application to four catchments in diverse climate regions. These are the Kalu Ganga (Sri Lanka), Pampanga, Angat and Kaliwa (Philippines), Yoshino (Japan), and Medjerda (Tunisia). The assumption of GCM stationary error in the future is verified by calibration in 1981–2000 and validation over 1961–1980 at two observation sites. The results show a favorable outcome. Overall performance of catchments was good for bias-corrected extreme events and inter-seasonal climatology, compared with observations. However, the suggested method has no intermediate grid scale between the GCM grid and observation points, and
requires a well-distributed observed rain gauge network to reproduce a reasonable rainfall distribution over a target basin. The results of holistic bias correction provide reliable, quantitative and qualitative information for basin or national scale integrated water resource management.

## 1 Introduction

To address climate change vulnerability in a projected period, global climate models (GCMs) are mandatory and reasonable
precipitation output is indispensible (Wood et al., 2004; Semenov and Doblas-Reyes, 2007). Policymakers are interested in basin or national scale impact assessment and mitigation (Klein et al., 1999), but GCM output has an improper grid scale for their needs (Solomon, 2007). GCMs have favorable continental-scale performance, but they lack major regional climate characteristics which are critical in national or watershed scale studies (Elshamy et al., 2009a). In addition, uncertainty of quantitative prediction may arise from the choice of GCM, unknown climatic forcing, or random variability of climate models





(Pielke Sr et al., 2012). To ascertain actual ranges of predicted change, most studies recommend multi-GCM simulations (Wilby et al., 2004; Paeth et al., 2008), but do not present clear, standardized criteria for GCM selection. Therefore, the present study introduces important benchmark and evaluation strategies for multi-GCM selection to reduce specific contingencies in basin-scale impact assessment.

A variety of statistical bias correction methods have been developed and some inter-comparisons have been carried out between these methods under various scopes (Teutschbein and Seibert, 2013; Watanabe et al., 2012; Lafon et al., 2013). From simple linear factor or delta change methods, multiplicative correction (Hay et al., 2000; Graham et al., 2007) to monthly nonlinear correction (Bordoy and Burlando, 2013), power laws or nonlinear corrections (Leander and Buishand, 2007; Hurkmans et al., 2010) have been developed. Some address only a mean difference error, whereas others cover both average and variance.

Others focus on correction of mean and standard deviation together with wet-day frequencies and intensity errors, which is called distribution mapping (Teutschbein and Seibert, 2013). From the statistical perspective, bias is corrected by a "transfer function" between the GCM and observed precipitation series. This is known as "quantile mapping", "histogram equalization or matching" or "probability distribution correction" (Wood et al., 2004; Piani et al., 2010a, b; Li et al., 2010; Maurer et al., 2013; Ines and Hansen, 2006). This approach has seen widespread use (Lafon et al., 2013; Schoetter et al., 2012; Piani and

Haerter, 2012; Argüeso et al., 2013) and shows superior performance to other methods, because high moment biases are effectively discarded (Teutschbein and Seibert, 2013). Because it is more straightforward and incurs less computational burden, this approach has become a popular and conventional tool for GCM bias correction. However, it requires consecutive decadal observed precipitation series to maintain time-invariant statistical properties.

The scope of the present study is determining a regional pattern of climate variability, which is certified by multi-GCM

selection, and ascertaining a detailed local fingerprint discrepancy through bias correction and spatial disaggregation. First, uncertainty of the multi-GCM is minimized by discarding imperfect GCMs as evaluated by past decadal regional climate feature generation, especially for precipitation. The major distinguished feature of the proposed method is its simultaneous action in addressing all GCM deficiencies, such as underestimation of extremes, poor simulation of inter-seasonal rain sequences, and wet day errors whereas most correction approaches treat only either one of them. This is achieved by mixing two probability

functions, one for tailed intensities that usually do not conform to a normal distribution and the other for monthly series of seasonal precipitation, including adjustment of wet day frequency errors. The major achievement is excluding biases in both amplitudes and frequencies of GCM precipitation time series on an interannual and yearly basis, which contributes to projection of basin-scale flood and drought analysis. Furthermore, the transferability of the application is confirmed by its implementation in three river basins in Asia and one in the Mediterranean region under diverse climates and local spatial variation as the bench-

mark of choice. A limitation of the approach is that reasonable numbers of well-scattered stations with a few decades of data are prerequisite. The assumption of time-invariant GCM bias during control and projection periods is verified by two decadal simulations, 1981–2000 as the calibration period and 1961–1980 as the evaluation period, at a few ground locations where long-term data are available. In this way, we developed a comprehensive and integrated statistical bias correction and spatial downscaling method together with tackling multi-GCM uncertainty. Particularly, analysis of potential impacts on basin-scale

precipitation is sucessfully achieved with low computation and high efficacy.





## 2   Study Area and Data

We acquired climate data from the Intergovernmental Panel on Climate Change (IPCC) Fourth Assessment Report (AR4) Coupled Model Inter-comparison Project 3 (CMIP3) under Special Report on Emission Scenarios 1 (SRES1). Total 24 GCM gridded daily precipitation data are accessed from the Data Integration and Analysis System (DIAS) server developed by the Earth Observation Data Integration and Fusion Research Initiative (EDITORIA) of The University of Tokyo. A 20-year control period (1981–2000) and projected future period (2046–2065) under the SRES A1B scenario were used. The pilot study sites were the Kalu Ganga River in Sri Lanka, Pampanga, Angat and Kaliwa watersheds in the Philippines, Yoshino River in Japan, and Medjerda River in Tunisia (Fig.1). Prioritizing applicability under heterogeneous climates, various regions were selected such as equatorial and tropical monsoon influence at the Kalu Ganga River, a tropical marine and equatorial monsoonal climate in the Philippines, a temperate humid climate at the Yoshino, and a semi-arid climate at the Medjerda River. Moreover, preference criteria included data availability, local orography and assorted geographical latitude, i.e., a strong land-sea thermal contrast basin (Kalu Ganga), northwestern Pacific island basin (Philippines), mountainous basin (Yoshino), and Mediterranean desert basin (Medjerda).

The Kalu Ganga River is in southwestern of Sri Lanka and empties into the Indian Ocean. Its basin is the second largest in Sri Lanka, covering 2766 km$^2$, and most area has heavy rainfall (annual average 4000 mm). A total of 26 stations were used, belonging to the Meteorology Department of Sri Lanka. Because of the hydrologic and topographical characteristics of the basin, its lower floodplain suffers from frequent floods during the southwest monsoon season (Ampitiyawatta and Guo, 2009). Accordingly, the demand for climate change studies is high for this area. The study area in the Philippines covers Metro Manila and adjoining areas, including the Angat (1,085 km$^2$), Kaliwa (including the Agos Basin; 280 km$^2$) and Pampanga (10,981 km$^2$) river basins. We used 21-station data from the Philippines Atmospheric, Geophysical and Astronomical Services Administration and Metropolitan Waterworks and Sewerage System.

The Yoshino River basin covers over 3700 km$^2$ on Shikoku Island located in the south-central of Japanese archipelago. The basin extends over four prefectures, Ehime, Kagawa, Kochi and Tokushima. More rain (> 3000 mm annually) in southern Shikoku results in flood damage, whereas low rainfall (1500 mm annually) in the northern part causes drought damage. The Ministry of Land, Infrastructure, Transport and Tourism of Japan furnished data from 25 rain gauge stations. The Medjerda, the major Tunisian river, originates in the semi-arid Atlas Mountains of eastern Algeria. Its catchment covers approximately 24,000 km$^2$, of which 7700 km$^2$ is in Algeria (Zielhofer et al., 2002). Our study covered only partial area of this catchment located within Tunisia. The catchment lies within a sub-humid to Mediterranean humid bio-climatic region. The rainy season is from September through May, with intense precipitation in autumn (Bouraoui et al., 2005). Precipitation data from 44 stations were provided by the government of Tunisia.



## 3 Methodology

### 3.1 Decrement of GCM uncertainty

To use GCMs with greater confidence and less uncertainty, model performance should be evaluated systematically (Perez et al., 2014). Using GCM global-scale simulated variables such as temperature, precipitation, radiation, and others comparison

to observations is the simplest technique to assess model performance. This kind of evaluation ensures the reliance on GCM projected results (Stocker et al., 2013). Thus, we established a unique benchmark and scoring system for multi-model selections. First of all, we examined model ability to capture major mesoscale and regional-scale climate features based on seasonal evolution in a synoptic scale domain, which influence basin-scale inter-seasonal precipitation. For examples, a pronounced Asia Pacific southwest monsoon system from May to November over Luzon Island in the Philippines (Murakami and Mat-

sumoto, 1994; Moron et al., 2009), strong seasonal frontal systems and typhoons stirring around the Yoshino River from June to November (Tachikawa et al., 2009), effects of the east Asian and south Asian monsoons and Tibetan Plateau high on the Kalu Ganga River, impacts of Mediterranean climate that include parts of Europe, and Mediterranean Sea and eastern Atlantic Ocean influences on the Medjerda River basin. Detailed information of local and regional domains, catchment areas, climate zones and number of stations is given in Table 1.

Next, climatological monthly long-term mean spatial correlation (Scorr;Eq. 1) and root mean square error (RMSE; Eq. 2) of precipitation, outgoing longwave radiation (OLR) in the local domain, plus sea level pressure, sea surface temperature (SST), air temperature, 850 hPa geopotential height, specific humidity, zonal wind and meridional wind in a regional domain from 1981 to 2000 were taken into account in model selection. All these data are available from DIAS. Global Precipitation Climatology Project (GPCP) precipitation (Adler et al., 2003), SST from the Hadley Centre (Rayner et al., 2003), OLR from

the National Oceanic and Atmospheric Administration (NOAA) (Liebmann, 1996), and other variables from the Japanese 25-year ReAnalysis (JRA-25) (Onogi et al., 2007) were considered as global references. Moreover, the criteria for precipitation were given the highest priority in GCM selection, indeed, seasonal and interannual variability in the study area should be reasonably simulated by the reliable GCMs. A comparison between 24 GCM and GPCP average precipitation in mm day$^{-1}$ from May–November over Luzon Island during the control period is shown in Fig. 2. This reveals the dissimilarity among

GCMs and why evaluation is a prerequisite for basin-scale impact study.

$$S_{corr} = \frac{\sum_{i=1}^{N} (x_i - \overline{x}) (y_i - \overline{y})}{\sqrt{\sum_{i=1}^{N} (x_i - \overline{x})^2 (y_i - \overline{y})^2}} \tag{1}$$

$$E_{RMS} = \sqrt{\frac{1}{N} \sum_{i=1}^{N} (x_i - y_i)^2} \tag{2}$$

where $x_i$ means GCM simulation, $\overline{x}$ refers to the zonal average of that simulation, $y_i$ is the reference data, $\overline{y}$ refers to the zonal mean of reference data, and $N$ is the total time series. The average of monthly Scorr and RMSE of all GCMs is used to



represent performance scores of each GCM. By comparing all analysis months' average Scorr and RMSE of each GCM to the average of all GCMs, the scoring scheme was implemented. A score of 1 was given to the GCM with larger spatial correlation and lower RMSE than the GCM mean. A score of 0 is assigned if either of the two conditions is satisfied; if neither are satisfied, a -1 is assigned. These scores were assigned to every key parameter, and we sum all scores of each GCM. According to the rank position of their total scores, it is clear which GCMs are skillful and which are undesirable for the target basins. This is the way to exclude the undesirable GCMs to reduce the ceratain uncertainty but not the way to choose the most excellent GCMs. The framework of multi-GCM selection and a schematic diagram of the scoring system for evaluating GCM performance are depicted in Fig. 3.

Table 2 shows the scores of all GCMs and their corresponding ranks according to the performance of all key parameters for Luzon Island. From Table 2, we excluded GCMs that did not have a precipitation score of 1 (i.e., higher Scorr and smaller RMSE than the mean of all GCMs) and GCMs with missing data. The same procedure was implemented for four catchments. Selected GCMs for each river basin are listed in Table 3.

## 3.2 Three-step statistical bias correction

Generally, GCM-simulated daily precipitation tends to occur more frequently but with less intensity than observed precipitation (Osborn and Hulme, 1998; Dai, 2006; Sun et al., 2006). It is well known that as a consequence, GCM precipitation without prior bias correction will provide unreliable impact assessment results (Feddersen and Andersen, 2005; Ines and Hansen, 2006; Sharma et al., 2007). In our study, three main GCM deficiencies were addressed, namely, underestimation of extremes, poor seasonal simulation, and high-frequency wet day error relative to observations, as shown in Fig. 4. We proposed a three-step statistical bias correction to eliminate these major GCM flaws.

### 3.2.1 Bias correction of extreme rainfall

Traditionally, extremes have been examined using the block maxima approach or annual maximum or minimum series (AMS) adjusted to the generalized extreme value, Gumbel, log-normal (LN3) and Pearson (P3) distributions (Gupta and Duckstein, 1975). The main drawback of AMS analysis is that it just considers the top as extremes in a year basic, in other words, second or third neglected peaks could be greater than other year?s top peak (Vicente-Serrano and Beguería-Portugués, 2003). We evaluated the performance of bias correction using 20 maxima (1981–2000) by defining extremes that were larger than the smallest one. After that, the best fit distribution from Gumbel or log-normal was identified, and bias corrected rain data was produced by inversing them. Another preliminary attempt was tested by partial duration series (PDS). The PDS approach includes all of maxima greater than the selected threshold, regardless of the year in which they occurred (Hershfield, 1973). To involve more records and improve estimation accuracy, a generalized Pareto distribution (GPD) function has been widely used (Hosking and Wallis, 1987; Smith, 1989; Lang et al., 1999).

Contrasts of the two methods were assessed at the Central Luzon State University (CLSU) station. A histogram compares observed data (dark blue) with raw Gfdl_cm2_0 GCM (light blue) and bias-corrected rainfall (red) (Fig. 5). There is a clear mismatch between CLSU and bias-corrected rain by the AMS near the 100 mm per day column in Fig. 5a, although there is a





reasonable matching in the PDS with GPD fitting (Fig. 5b). This is because all rain data greater than the lowest of 20 maxima are defined as extremes in the AMS method, but the lowest one could be smaller than the normal rain of the other year. Moreover, excessive numbers of extremes in the GCM than CLSU degrade bias correction efficiency. Therefore, the GPD function is desirable for extreme rainfall bias correction, and many recent studies have demonstrated the superior performance of the

5 GPD distribution for extreme hydrologic variables using PDSs (Madsen et al., 1997) such as annual maximum flood (Hosking and Wallis, 1987), maximum dry spell (Vicente-Serrano and Beguería-Portugués, 2003), extreme wind speed (Holmes and Moriarty, 1999), and extreme precipitation studies (Madsen et al., 1994). The advantage of GPD with PDS is that it disregards small precipitation values and focuses only on high intensity series. More specifically, rainfall exceedances, i.e., rain amounts greater than a given threshold or series of peaks exceeding an upper limit (Dargahi-Noubary, 1989) can be approximated by

10 GPD if the threshold and number of observations are sufficiently large. These exceedances are typically assumed to have a GPD $(k, \alpha)$ whose cumulative distribution function (CDF) is defined by

$$F(x; k, \alpha) \begin{cases} 1 - \left[ 1 - \frac{\kappa(x-\xi)}{\alpha} \right]^{\frac{1}{\kappa}}, & \kappa \neq 0, \ \alpha > 0 \\ 1 - \exp\left( \frac{-(x-\xi)}{\alpha} \right), & \kappa = 0, \ \alpha > 0 \end{cases} \tag{3}$$

Here, $\kappa$ and $\alpha$ are shape and scale parameters. The range of $x$ is $x > 0$ for $\kappa \leq 0$ and $0 < x < \kappa$ for $\kappa > 0$. The case $\kappa = 0$, which is the exponential distribution, is the limiting distribution as $\kappa \to 0$ (Pickands, 1975). Several methods have been proposed for

estimating the parameters of the GPD, such as the maximum likelihood (ML), methods of moments (MOM) and probability weighted moments (PWM) estimators (Hosking and Wallis, 1987). The heavy-tailed distributions are the most common in hydrology. MOM estimation is a more efficient T-year event estimator than ML and PWM (Madsen et al., 1997). We selected MOM estimators, and the shape and scale parameters are given by

$$\kappa = \frac{1}{2} \left[ \frac{\mu^2}{\sigma^2} - 1 \right] \tag{4}$$

$$\alpha = \frac{1}{2} \mu \left[ \frac{\mu^2}{\sigma^2} + 1 \right] \tag{5}$$

Here, $\mu$ is the sample mean and $\sigma^2$ the sample variance (Hosking and Wallis, 1987). The extreme event $X_T$ in a period of $T$ years is obtained using

$$X_T = \xi + \frac{\alpha}{\kappa} \left[ 1 - \left( \frac{1}{\lambda T} \right)^{\kappa} \right], \qquad \kappa \neq 0 \tag{6}$$

where $\lambda$ is the average number of annual events greater than the threshold. A major issue using PDS is selection of the lower

bound $\xi$, where the tail begins. This value should be sufficiently small to ensure inclusion of as much relevant information as possible (Lin, 2003). Various methods have been proposed to determine the most appropriate lower bound (Madsen et al.,



1997; Rosbjerg et al., 1992). The most popular methods for estimating the threshold are graphical methods, Hill plots, and mean excess function (MEF) plot. Hill (1975) proposed the following estimator for $\kappa$:

$$\kappa = \frac{1}{a-1} \sum_{i=1}^{a-1} \ln X_{i,N} - \ln X_{a,N}, \qquad a \geq 2 \tag{7}$$

where $a$ is upper-order statistics (number of exceedances), $N$ is sample size, and $X_{i,N}$ represents $i^{th}$ order statistics of random variable $X$. A Hill plot can be constructed such that the estimated $\kappa$ is plotted as a function either of $a$ (from Eq. 7), upper-order statistics, or the threshold $\xi$ as in Fig. 6a. A threshold should be selected from the plot where the shape parameter $\kappa$ is relatively stable in the Hill estimator (Embrechts et al., 1997). The MEF is also a popular tool to assist the choice of $\xi$, and is defined by

$$e_n(\xi) = \frac{\sum_{i=1}^{n} (X_i - \xi) I_{(X_i > \xi)}}{\sum_{i=1}^{n} I_{(X_i > \xi)}}, \qquad \xi \geq 0 \tag{8}$$

where $X$ denotes the random variable. $I = 1$ if $X_i > \xi$, and is $0$ otherwise. $e_n(\xi)$ is the empirical MEF of $\xi$, and is the sum of the exceedances of the threshold $\xi$ divided by the number of data points exceeding the threshold $\xi$, as in Fig. 6b. In MEF, the suitable lower bound is the point of sudden change of slope of the graph. Precipitation intensities equal or greater than the 95th or 99th percentile are considered extremes. The choice of those two percentiles mainly depends on the total number of wet days. In this way, the GPD threshold is defined by using descending series of rainfall at each ground station. From the descending order of the corresponding GCM gridded data, the equal number of extreme events is determined for bias-correction. Here, it is assumed that GCM uncorrected extreme series follow the same GPD distribution as the ground station. However, the smallest RMSE between corrected GCM extremes and observed data is tuned for the best fit. Finally, bias-corrected GCM extremes are calculated using

$$x'_{\text{Gcm}} = F_{\text{Obs}}^{-1}(F_{\text{Gcm\_Past}}(x)) \tag{9}$$

Here, $x'_{\text{Gcm}}$ represents bias-corrected extreme precipitation, $F_{\text{Obs}}^{-1}$ is the inverse GPD function of observed extremes, and $F_{\text{Gcm\_Past}}(x)$ is the GPD function of GCM gridded data during the control period.

### 3.2.2 Frequency of wet-day error

There are errors in the frequency of wet days in GCM because of inadequate parameterization of several processes associated with cloud information (Elshamy et al., 2009b). We solved this problem by using simple rank-order statistics of the entire time series. In descending order of GCM rain amount, GCMs have low-intensity rainfall with a large number of rain days at the end. By assuming the same number of rain days in observation as in GCM output, we define the GCM rain which is the same rank of the last rain from observation as the threshold for correction. Beyond that threshold, GCM rainfall is changed to zero for



correction. The same threshold is used in the future projection. The reason is that the future is unknown and to investigate the difference in the total number of dry days and the change of continuous dry or well spell in the near future. However, we should be aware that this correction method ignores the continuity of rain or dry-day error and solely considers the total frequency error.

### 3.2.3 Bias correction of inter-seasonal precipitation

The final component of correction is for the bias of seasonal precipitation. All rain series between the two thresholds (extreme and total rain day correction) are defined as normal rain for both GCM and rain gauge data. A two-parameter gamma distribution was used for bias correction. We assumed that the CDF of monthly normal daily rainfall at a certain grid point follows the gamma distribution function. The shape and scale parameters for all 12 calendar months were estimated by the simple MOM. Then, all monthly grouped precipitation was adjusted for fitting the gamma distribution by mapping the CDF of daily GCM precipitation to that of observed precipitation. The two-parameter gamma distribution is defined by

$$F\left(x;\alpha,\beta\right) = \frac{1}{\beta^{\alpha}\Gamma\left(\alpha\right)}x^{\alpha-1}\exp\left(-\frac{x}{\beta}\right) \tag{10}$$

Here, $\alpha$ and $\beta$ are shape and scale parameters of the gamma distribution determined by the MOM estimation, and as in Eq. 4 and Eq. 5. The corrected normal rainfall data are calculated by the fitted gamma CDF using Eq. 9. Using projected GCM precipitation (2046–2065), bias was corrected by the same transfer function between GCM historical and observation data, via

$$x'_{\text{Gcm\_Fut}} = F^{-1}_{\text{Obs}}(F_{\text{Gcm\_Past}}(x_{\text{Gcm\_Fut}})) \tag{11}$$

Here, $x'_{\text{Gcm\_Fut}}$ represents bias-corrected extremes for projected GCM data, $F_{\text{Gcm\_Past}}(x_{\text{Gcm\_Fut}})$ is the same function for GCM hindcast fitting by projected precipitation ($x_{\text{Gcm\_Fut}}$), meaning that biases in the present climate are assumed stationary. In this way, model precipitation bias was corrected by the GPD and gamma distribution, including both the frequency and intensity error simultaneously.

### 3.3 Exceptional seasonal extreme bias correction

In the three-step bias correction, the GPD attempts to just get rid of bias from the top-ranking rain days of the 20 years. Particularly, interannual extreme events in the seasonal or monthly distributions are ignored. This disproportional ratio of numbers of extremes in month by month between observations and GCM may cause inconsistency of inter-seasonal variation. Given the substantial contribution of extremes to monthly means, deviation of the inter-seasonal pattern is obvious at Sameura station (Yoshino River) (Fig. 7b). The blue thick line represents the observation, various colors for each GCM, gray area for the range of GCM rainfall, and red line for the GCM mean. After bias correction, the Baiu season had larger amplitudes than observed. There is a sudden decline before the typhoon season, and then a smaller peak forms again in September. However, the actual condition shows a continuous increase of average rainfall from the Baiu season through the end of the typhoon





season. Therefore, the exceptional seasonal extreme bias correction was implemented to solve this problem. The source of error is examined by the monthly distribution of extreme and normal rainfall, (Fig. 8). This reveals that the monthly normal rainfall bias correction was perfect (Fig. 8a) and bias was mainly contributed by extreme seasonal variation, as compared with ground stations (Fig. 8b). This is a weak point of the statistical bias correction, which does not consider correlated timing

of extremes. Most of the GCMs did not simulate the considerable extreme amounts during the late wet season (August–November), magnified heavy rainfall in early May and June, and then there was a large drop in July (Fig. 8 (b)). On the contrary, Sameura data (blue thick line) show no extremes in May and then dramatically increase in total seasonal rainfall in June and July. The peak is in August and September, mainly influenced by tropical cyclones. This may be the same finding as Kang et al. (2002), in which most of the GCMs showed the Baiu-Meiyu rainband peak phase appeared about one month

earlier than observed. It is evident that most GCMs missed the extratropical rain that greatly affects the rainy season in East Asia; this appeared in the observations until the end of August. Most GCMs failed to simulate typhoon events which was the mature stage of tropical cyclones usually occuring with synoptic-scale disturbances. There was a positive bias in early June and a negative bias at the end of July. This is similar to the finding that the total number of tropical storms in GCM output is underestimated compared with the frequencies of observed typhoons (Matsuura et al., 1999).

For the above reason, the exceptional case of seasonal extreme bias correction was necessary for the study area. Various trial pairs of seasonal bias correction were carried out. The subtropical monsoon season of June–July (JJ), typhoon season of August–September (AS), and remaining combinations had better outcomes than the other trials, but not the perfect one. Finally, the mean frequency of extremes during June through July and August through September worked very well (Fig. 15f). This means that not only the general method of statistical bias correction but also a seasonal bias correction should be executed

when the precipitation is strongly influenced by very heavy rain such as from tropical storms or typhoons.

### 3.4   Evaluation of stationary bias correction function

Average GCM bias is statistically the same between two sets of years (Maurer et al., 2013), and we assumed constant model biases in the projected period. To ensure the adequacy of the stationary bias assumption over decadal timescales and boost reliability in the future projection, we applied the bias correction method to a different set of years. That is, for a calibration

period of 1981–2000 and evaluation period 1961–1980, for Matsuyama station on Shikoku Island (Japan) and Oued Mellegue station in the Medjreda Basin (Tunisia).

  Box plots of 25th, 50th and 75th percentile extreme rainfall from observation and bias-corrected GCMs for 1981–2000 (a and b) and 1961–1980 (c and d) are presented in Fig. 9 for Matsuyama and Fig. 10 for Oued Mellegue. In both figures, label ?a? on the x-axis is for observation and b, c, d, e, f, g on that axis refers to the selected GCMs from Table 3. The third column of

that table is for Matsuyama and the last column for Oued Mellegue. Figures 9b and 10b show strong matching of interquartile range between all box plots and in the evaluation period (Figs. 9d and 10 d), especially the mean and lower quartiles, compared with uncorrected GCM rain in both periods (Figs. 9a and c) and 10a and c). Moreover, upper quartiles and interquartile range variation were also similar to observation during the calibration and evaluation period after correction. The outliers of GCMs became equal to the outliers of observation "a" during 1981–2000 after correction, although the original GCM outliers were





randomly smaller than observation. Outlier correction in the evaluation period was not as effective as in the calibration period, at both stations. Nevertheless, the stationary error assumption is reliable and the corresponding GPD transfer function during calibration was unambiguously qualified for the evaluation period.

Long-term monthly precipitation variations over both decadal scales are illustrated in Figs. 11 and 12. Here, a blue thick
line refers to the observed monthly climatology of precipitation, a red thick line to the ensemble means of all GCMs and other color lines for each GCM. In Figs. 11 and 12, panels a and c are for uncorrected GCM precipitation and panels b and d for corrected GCMs. Bias in the uncorrected GCMs' range (gray area) is obviously broader than that for corrected GCMs at both stations. Although the original GCM precipitation had no bimodal rain peak at Matsuyama (Fig. 11 a and c), the bias-corrected outcome shows a similar pattern as with the ground station during the calibration. The corrected GCM monthly means over
the evaluation period had a reasonable outcome but a slightly broader range than over the calibration period at both stations. Bias-corrected ensemble GCM means performed better for both time periods and stations. In brief, the proposed bias correction satisfactorily eliminated GCM bias for both tailed extremes and seasonal climatological rainfall.

## 4 Results

### 4.1 Correction of wet-day frequency error

Wet-day frequency error was corrected for only the total number of wet days over 20 years, and we did not consider discrepancies in either the monthly scale or continuous wet days. Consequently, the continuity of wet days and consecutive dry spells might differ from the observed frequency of wet and dry days. Figure 13 is an example comparison of 20-year total maximum continuous wet days between bias-corrected GCMs and Sameura station (Yoshino River) during 1981–2000. The full horizontal bar length specifies the total maximum continuous wet days in each GCM, and various color segments indicate
each year. It is clear that the ingv_echam4, csiro_mk3_0 and csiro_mk3_5 GCMs have longer maximum wet-day continuity than Sameura, and the other three GCMs were nearly the same during the control period after bias correction. Year-by-year inequality can be judeged by the width of the bar. Consequently, a loss of continuity of wet days may adversely affect flood- or drought-related hydrologic simulation. The mean of maximum consecutive wet days during the 20 years was 9.3 for Sameura, 16 for the csiro_mk3_0, 12.65 for csiro_mk3_5, 17 for ingv_echam4, 10.05 for iap_fgoals1_0_g, 9 for miroc3_2hires, and 8.95
for miroc3_2medres GCMs (not shown). At the same time, mismatch of continuous dry spells may affect drought analysis for the catchment.

### 4.2 Extreme rainfall bias correction

The bias-corrected GCM extremes compared with in situ station data in the ranking order statistics are shown in Fig. 14. Panels a and b are for Halwataru station of the Kalu Ganga river, c and d are for Central Luzon State University (CLSU) station of
30 the Pampanga river, e and f are for Sameura station in the Yoshino Basin, and g and h are for Ain Beya Qued station of the Medjerda River. It can be seen easily that the raw GCM extremes in descending order underestimate relative to the ground





station (Fig. 14a, c, e and g) which is the common bias of undervalues from GCM resolution. However, the bias of each rank was effectively removed by GPD fitting, especially for the uppermost ranks (Fig. 14b, d, f and h). In particular, the performance may depend on the difference between the first five in the uppermost series, because if their gap are large, the GPD may deviate from the fit in the upper ranks. However, Fig. 14 shows only one point from each river basin; there may be some points that
cannot conform to the GPD fit well because of limited data entries in the GPD series.

## 4.3   Inter-seasonal rainfall bias correction

The final evaluation is for bias-corrected inter-seasonal rainfall. Figure 15a, c, e and g shows monthly average rainfall during the control period before bias correction, and Fig. 15 (b, d, f, h) shows the result after bias correction. A large bias range (gray area) can be found at all stations as well as rain intensity underestimates over the entire wet season, except for CLSU before
correction. There is no bimodal seasonal variation (Fig. 15a) in the original GCM outputs for Halwataru station in the Kalu Ganga Basin. This may be associated with the poor representation of the topography effect of Sri Lanka Island on the ocean and failure to simulate the bimodal monsoon precipitation in the aforementioned basin. However, bias was effectively removed (Fig. 15b). The mean of the GCM ensemble (red thick line) represents performance of all selected GCM bias correction at that location. There was small underestimation in the southwest monsoon season and a bit overestimation in the northeast monsoon
season. This bias may come originate from unequal frequencies of extreme events during the two monsoon seasons compared with the ground station data. For CLSU station, Fig. 15c and d shows that the GCM ensemble simulation nearly fit the monthly mean of the station data, owing to powerful achievement of the bias correction method. After the special seasonal extreme and normal rainfall bias correction, a comparable match of inter-seasonal pattern was achieved at Sameura station, as shown in Fig. 15e and f. The result demonstrates the success of the special seasonal correction method. A comparison of interannual
seasonal variability of GCM rainfall before and after correction at Ain Beya Qued station (Medjerda River) is shown in Fig. 15g and h. Undervalued bias is noticeable over the entire season and most GCMs did not represent the prominent signal mode seasonal rain pattern. After bias correction, all GCMs changed in the same manner as station observations, revealing that biases were effectively corrected. This clearly demonstrates the superior performance of the bias correction method for long-term inter-seasonal variation at point scale.
We also evaluated the basin?level, long-term climatological rainfall spatial distribution during the control period. In the pilot study areas, there are enough rain gauges to produce a reasonable spatial distribution. If the bias correction performs satisfactorily at all ground stations, bias-corrected interpolated surfaces over the target basin should be comparable to the surface climatology pattern from those stations. Therefore, bias-corrected GCM precipitation from all rain gauges were interpolated to the basin scale using spline surface interpolation in ArcGIS software. It was then easy to compare the distribution shape
and intensity at basin scale simultaneously. The mean of the selected ensemble GCM spatial pattern was prepared for all four basins.

      In the Kalu Ganga River basin, May climatology was chosen, because it is one of the heaviest rainfall months during the rainy season. There is a clear division between a high precipitation zone in the northwest or downstream of the basin and a low precipitation zone in the southeast or upper portion of the basin. The original GCM average precipitation in May shows





no variation in the entire basin (not shown) and consistent underestimation across the basin before correction, and the GCM rainfall distribution is comparable to the observed after correction (Fig. 16b). The absolute error is $< 100$ mm month$^{-1}$, and error is large in the high intensity area and small in the low intensity area. Fig. 17a and b compares the ground station and bias-corrected GCM August precipitation distribution in the Pampanga, Angat and Kaliwa watersheds. The GCM average showed

excellent performance at every location, and as a result, the distribution was nearly identical to the observed. Nevertheless, the interpolation between the irregularly spaced observations increased the absolute error to $> 100$ mm month$^{-1}$ in the western part, and there was underestimation by 200 mm month$^{-1}$ in the eastern area. The error range of all ground stations is about - 70 to 50 mm month$^{-1}$.

For the Yoshino River basin, September climatological precipitation was selected, because September is one of the highest

precipitation months in the seasonal cycle of the year. The interpolated precipitation surface for September is portrayed in Fig. 18a for the observation and Fig. 18b for post-correction. Before bias correction, all GCMs showed the same rain intensity over the entire basin. This may be because the basin is covered by only one or two GCM grids. The corrected result matched well the various intensity groups, and bias of all stations was effectively corrected for both intensity and pattern variation during the past simulation. However, there was a small absolute error around - 5 to - 20 mm month$^{-1}$. In the Medjerda River basin, the

44 gauges are fairly evenly distributed over the catchment area, except outside of Tunisia. We assumed two stations in Algeria to cover the entire Medjerda Basin for the spatial distribution and interpolation. A comparison between the long-term mean January rainfall at the ground stations and bias-corrected GCM rainfall is shown in Fig. 19a and b, respectively. The corrected pattern shows an underestimation area in the northwest of the basin, and overestimation in the uppermost part. This inaccuracy may be attributed from the limited points for interpolation. Therefore, in reducing the inevitable uncertainties of bias-corrected

GCM output and producing more reliable projected outcomes, the proposed statistical bias correction method was verified at both point scale and basin level for all types of rain especially extremes and inter-seasonal variation in continuous time series. In summary, the basin-wide simulation performed well after bias correction and it is reasonable for use in future impact assessment studies.

### 4.4 Climate change scenario results

Through the descriptive and inclusive scheme of bias correction, a certain amount of uncertainty from GCM bias was discarded. Hence, factual understanding of future scenario change could be examined quantitatively. Fig. 20 depicts absolute change of 99th-percentile rainfall (excluding 0 mmday$^{-1}$) from selected models during the projected period, for various river basins. That change was determined from the difference between the 99th percentile of bias-corrected projected rainfall and observed values derived from the average of all stations in each model. Error bars in Fig. 20 represent the standard deviation among

stations within each watershed.

The Medjerda desert basin in a semiarid region had the smallest change of all GCMs compared with the ground station average of 45 mm day$^{-1}$. The other three catchments showed a remarkable increase in most of the GCMs. Among these, the Kalu Ganga basin showed that a roughly average rainfall intensity of 20 mm day$^{-1}$ may increase relative to the present mean of 110 mm day$^{-1}$. The Yoshino River received the heaviest rainfall (173 mm day$^{-1}$) among all basins during 1981?-2000.





Most GCMs agreed in showing an increasing trend of extremes, but one showed a major decrease in change. Finally, rain will intensify by 17 mm day$^{-1}$ from the GCM mean. However, the range of intensification looked disparate across various in the Yoshino Basin. Watersheds in the Philippines will receive more heavy rain, with a nearly 20 mm day$^{-1}$ surplus over the model average, although one GCM output a small negative trend. The ground station mean is 105 mm day$^{-1}$ in the present decade, and it needs to be well prepared for 20% additional heavy rain.

Figure 21 summarizes the relative change of monthly climatological rainfall in each river basin, which is defined as the ratio between changes in projected monthly climatological rain divided by the ground station values. First, this monthly climatological change was calculated for all stations by each GCM, then the average of all points was taken, and finally the long-term mean of the all-GCM average was determined for every calendar month. Monthly error bars in Fig. 21 refer to the standard deviation or spreading range among projected changes of each GCM, which was calculated from the mean of all stations within a target basin. Relative change of monthly climatological rain in the Yoshino Basin showed too little increase in the August–September rainy season and a substantial increase in May, indicating that Baiu frontal rain will be stronger. In addition, August to September has the most rain on an annual basis and so a slight increment during these 2 months can adversely affect the river basin flow regime. A contradictory direction of change for the bimodal rainy season was evident for the Kalu Ganga basin. Surprisingly, almost all of the first half of the rainy season trend decreased a minimum of -0.15 times the present amount, and the secondary peak rainy season had a significant increasing trend by nearly a factor of 0.2–0.4. In the Pampanga, Angat and Kaliwa watershed area, the largest increases were for the dry season (December through March), and there was too little increase for the major wet season from the ensemble model average projection. Moreover, uncertainty ranges between models (represented by the error bars of each month) were especially large in the dry season, with little variation in the wet season. In contrast with the other three basins, all monthly change statistics were negative in the Medjerda Basin. A much greater negative change was obvious in the wet season from November through March. Furthermore, there was a similar reduction of total rainfall in the dry season, and the consequences will become an important issue for water resources in the Medjerda Basin.

## 5 Summary and conclusions

This paper proposes a simple, comprehensive and integrated bias correction scheme combining two statistical approaches, one for seasonal normal rainfall and the other for extreme or tailed rainfall. These address the main GCM deficiencies such as the underestimation of extreme rainfall, too much rain dary error of near-zero drizzle rainfall, and poor seasonal or inter-annual variation relative to observations at basin scale. Furthermore, a multi-model selection method was proposed that uses regional heterogeneity effects on local variation as standard practice. This eliminates uncertainty from inefficient GCMs in basin-scale simulation. The selection method is executed by spatial pattern analysis and quantitative RMSEs of atmospheric variables for a region, which mainly contribute to the wet season in the target basin. After multi-model selection, a three-step bias correction composed of GPD for extreme rainfall, ranking order statistics for frequency error, and gamma distribution for monthly normal rainfall was implemented for all stations and their corresponding GCM grids. The main advantage is that bias




in both intensity and frequency of GCM precipitation are sufficiently eliminated on an inter-seasonal or yearly basis, which is crucial to estimate renewable freshwater resources and the impacts of flood and drought. This encourages the application of this method to multiple river basins under different climates worldwide.

The evaluation of bias correction was figured out by descending order statistics, various return periods, box plots of the 25th, 50th and 75th percentiles for extreme rainfall, and climatological monthly variation patterns. The overall assessment was done for both station and catchment level. Bias-corrected results at both basin and point scale reveal the effective elimination of bias, but there were some exceptions such as the seasonal extreme bias correction for the Yoshino Basin. Conventionally, the statistical bias correction has no temporal reference and just reshapes the required statistical properties of variables. We found that this shortcoming generates undesired bias in the timings of extremes at seasonal scale compared with observation, and reveals GCM weakness in simulating smaller numbers of tropical typhoons in the Yoshino basin, where most heavy rainfall is introduced by typhoons. This obstacle was overcome by various trial–and–error approaches of alternative pairs of months in the seasonal extreme bias correction.

Another important point is that total wet days at the stations was used in model wet-day frequency error correction. This may result in the loss of rainfall continuity, such that continuous dry spells or wet days may influence the simulation of river flow routing, especially in very dry or very wet basins. The common assumption of time-invariant bias transferability was also certified using two different decadal series at two stations, one in Japan and the other in Tunisia. This was done because of the availability of long-term data. Comparison of these two decadal series demonstrated the proficiency of the statistical bias correction, boosting confidence of its use in future projections.

One of the obligations of the proposed method is long-term precipitation datasets from well-distributed, in situ rain gauge networks covering a basin. This is compulsory to obtain a reasonable spatial distribution of precipitation, which can influence auxiliary impact studies. Another requirement is the determination of the number of extremes at a certain percentile, such as 99th or 95th percentile of wet days for the sense of extremes, especially for semiarid basins such as the Medjerda. In this basin, the proportion of wet days is much smaller than that of dry days. Thus, there must be some balanced adjustment for adequate members in the GPD fitting and the exclusion of some median- and low-intensity rain days after the wet-day frequency correction. This is because the introduction of median rain days as extremes may degrade GPD fitting of the top ranks. By examining the various responses of GCM precipitation error as compared with the local basin scale, this study accomplished the comprehensive statistical bias correction, thereby reflecting realistic precipitation intensities for catchment-scale impact studies.

*Acknowledgements.* This research was carried out as part of the Research Program on Climate Change Adaptation (RECCA) and Data Integration and Analysis System (DIAS) project 2011–2015. We are grateful to the EDITORIA science team for permission to use the GCM time series dataset of the DIAS project. The authors thank all the data providers.



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

**Table 1.** List of catchments for GCM selection method, including catchment area in square kilometers, climate zone, number of stations, regional domain, and local domain

| Catchments | Area($km^2$) | Climate zone | Stations | Regional Domain | Local Domain |
|---|---|---|---|---|---|
| Kalu Ganga | 2,179 | tropical monsoon | 26 | 50 °E - 150 °E, 0 ° - 41 °N | 60 °E - 110 °E, 10 °N - 41 °N |
| Pampanga & Angat, Kaliwa | 12,350 | tropical marine | 21 | 80 °E - 160 °E, 0 ° - 20 °N | 115 °E - 130 °E, 10 °N - 20 °N |
| Yoshino | 3,700 | temperate and humid | 25 | 80 °E - 160 °E, 5 °N - 61 °N | 32 °E - 135 °E, 32 °N - 36 °N |
| Medjerda | 24,000 | semi arid | 44 | 30 °W - 50 °E, 20 °N - 51 °N | 5 °E - 15 °E, 25 °N - 40 °N |





**Table 2.** Total scores of GCMs for Luzon island in the Philippines

| Rank | GCM | Scores |
|------|-----|--------|
| 1 | gfdl_cm2_1 | 7 |
| 2 | gfdl_cm2_0 | 6 |
| 3 | mpi_echam5 | 6 |
| 4 | cccma_cgcm3_1 | 5 |
| 5 | cccma_cgcm3_1_t63 | 5 |
| 6 | ingv_echam4 | 5 |
| 7 | ipsl_cm4 | 5 |
| 8 | ncar_ccsm3_0 | 4 |
| 9 | bccr_bcm2_0 | 3 |
| 10 | giss_aom | 2 |
| 11 | miub_echo_g | 2 |
| 12 | mri_cgcm2_3_2a | 2 |
| 13 | csiro_mk3_0 | 1 |
| 14 | csiro_mk3_5 | 1 |
| 15 | miroc3_2medres | 1 |
| 16 | ukmo_hadgem1 | 1 |
| 17 | miroc3_2hires | 0 |
| 18 | giss_model_e_r | -1 |
| 19 | ukmo_hadcm3 | -1 |
| 20 | cnrm_cm3 | -2 |
| 21 | iap_fgoals1_0_g | -3 |
| 22 | inmcm3_0 | -4 |
| 23 | giss_model_e_h | -5 |
| 24 | ncar_pcm1 | -6 |





**Table 3.** Selected GCMs for Kalu Ganga River basin in Sri Lanka, Yoshino Basin in Japan, Pampanga, Angat and Kaliwa basins in the Philippines, and Medjerda Basin in Tunisia

|   | Kalu Ganga | Pampanga Angat and Kaliwa | Yoshino | Medjerda |
|---|---|---|---|---|
| a | gfdl_cm2_1 | gfdl_cm2_0 | csiro_mk3_0 | cccma_cgcm3_1 |
| b | cccma_cgcm3_1 | gfdl_cm2_1 | csiro_mk3_5 | cccma_cgcm3_1_t63 |
| c | cccma_cgcm3_1_t63 | ingv_echam4 | ingv_echam4 | miroc3_2hires |
| d | gfdl_cm2_0 | ipsl_cm4 | iap_fgoals1_0_g | mpi_echam5 |
| e | miroc3_2mederes | csiro_mk3_0 | miroc3_2hires | mri_cgcm2_3_2a |
| f | mpi_echam5 | miroc3_2mederes | miroc3_2mederes | ingv_echam4 |
| g | ingv_echam4 |  |  |  |
| h | miroc3_2hires |  |  |  |

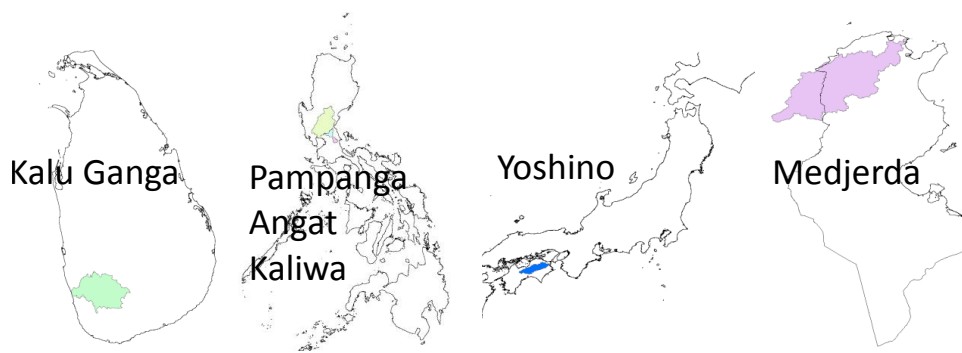

**Figure 1.** Locations of four pilot study sites: Kalu Ganga Basin in Sri Lanka; Pampanga, Angat and Kaliwa basins in the Philippines; Yoshino Basin in Japan; Medjerda Basin in Tunisia





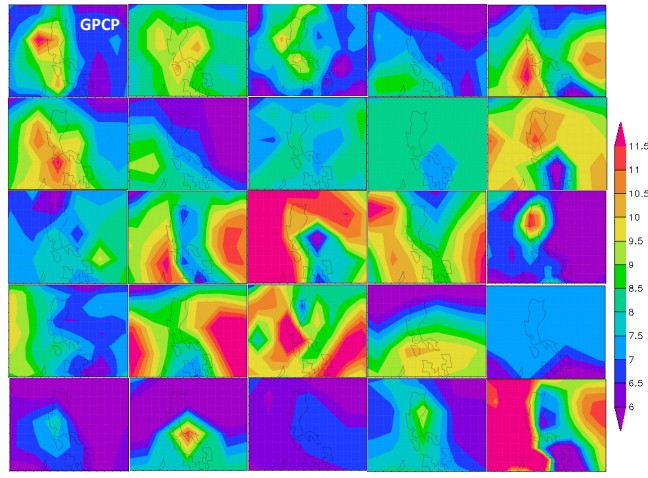

**Figure 2.** Comparison of GPCP and 24 GCMs mean precipitation from May to November over the Luzon Island in the Philippines during 1981–2000

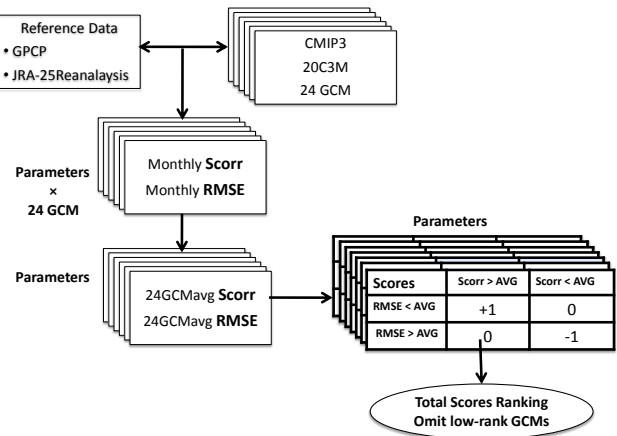

**Figure 3.** Schematic diagram of multi-GCMs selection framework





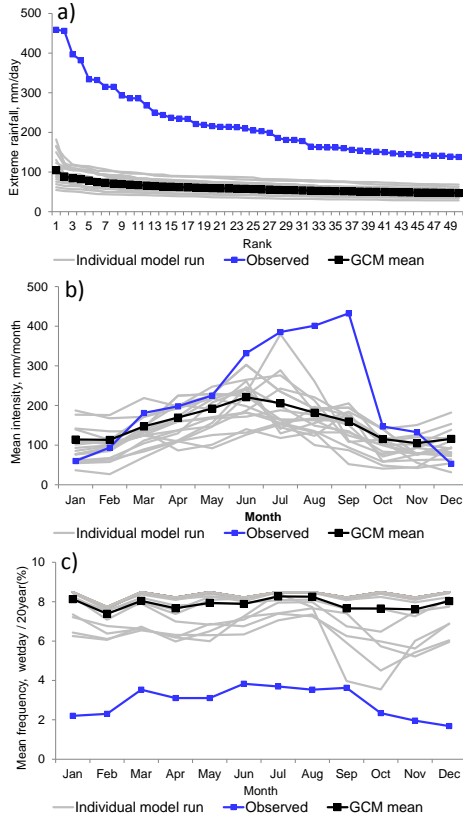

**Figure 4.** (a) Underestimation of extremes; (b) poor seasonal representation; (c) large wet-day frequency error of GCM compared with Sameura station on the Yoshino river during 1981–2000

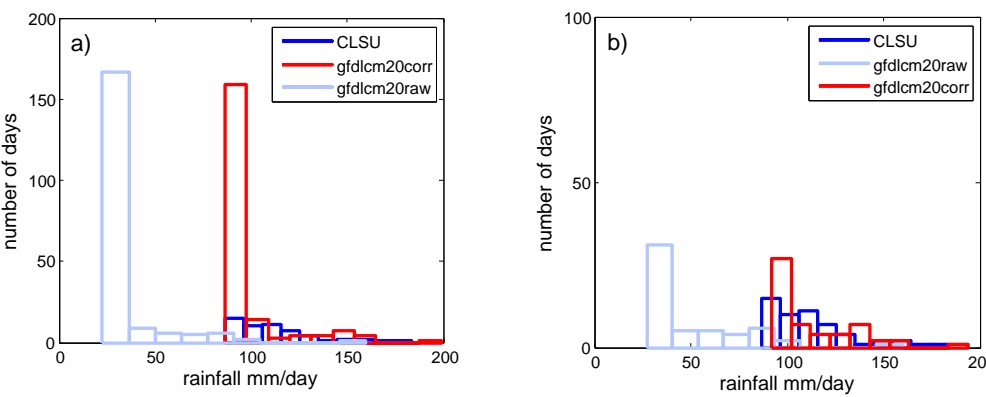

**Figure 5.** Histogram of Gfdl_cm2_0 GCM rainfall distribution before bias correction (light blue line), after bias correction (red line), and observation (dark blue line) at Central Luzon State University (CLSU) station, by (a) AMS series fitting and (b) GPD series fitting



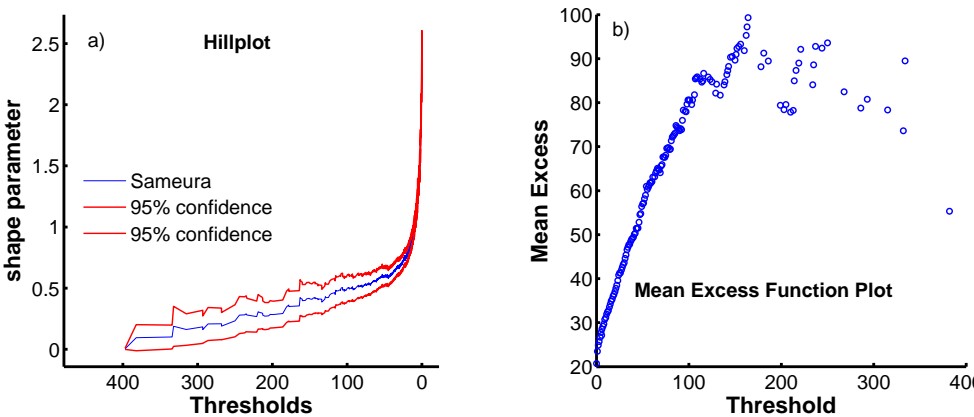

**Figure 6.** (a) Hill plot and (b) mean excess function (MEF) plot for Sameura station on the Yoshino River in Japan

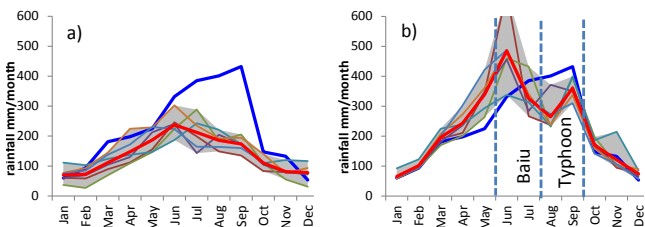

**Figure 7.** Inter-seasonal variation at Sameura station during 1981–2000: (a) original GCM data; (b) bias in Baiu and typhoon season after three-step bias correction

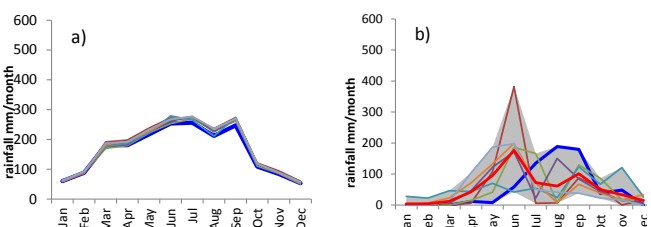

**Figure 8.** Error checking for (a) monthly normal rainfall and (b) monthly extreme rainfall




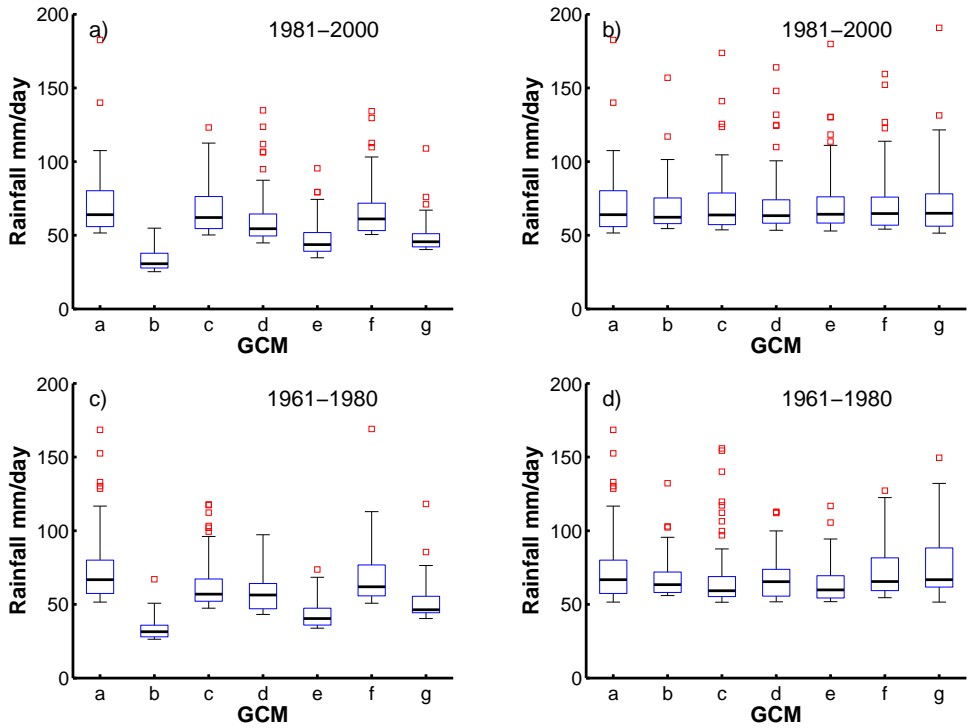

**Figure 9.** Box plot of extremes (a, c) before bias correction and (b, d) after correction for calibration period 1981–2000 and evaluation period 1961–1980 at Matsuyama station (Japan) (outliers beyond ±2 times the interquartile range shown by red squares)

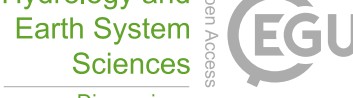

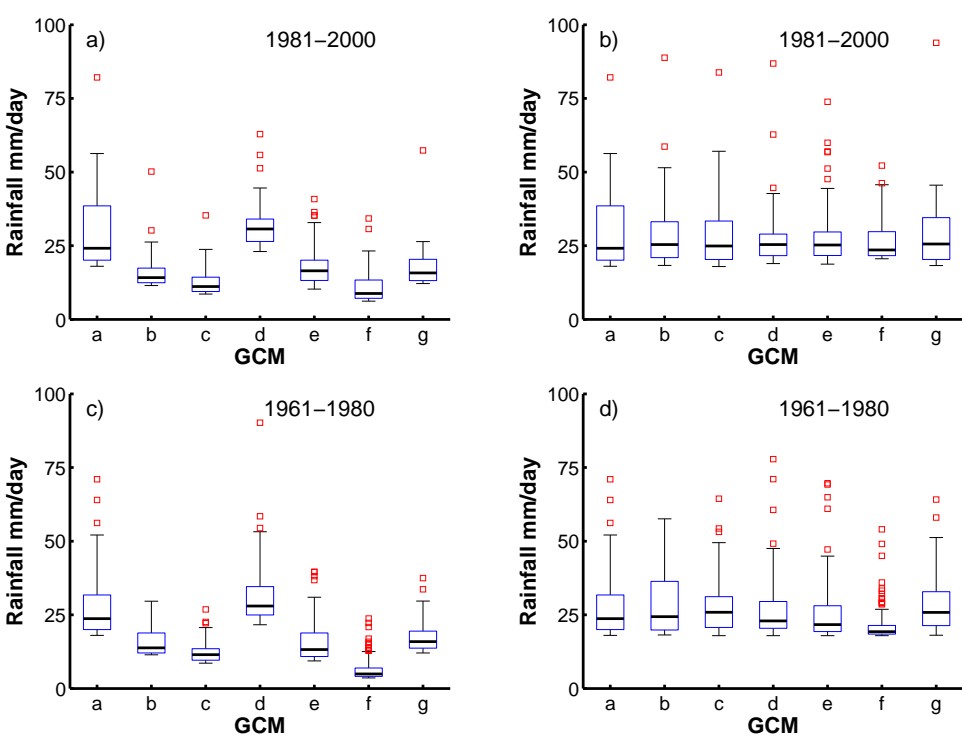

**Figure 10.** Box plot of extremes (a, c) before bias correction and (b, d) after correction for the calibration period 1981–2000 and evaluation period 1961–1980 at Oued Mellegue station (Tunisia) (outliers beyond ±2 times the interquartile range shown by red squares)





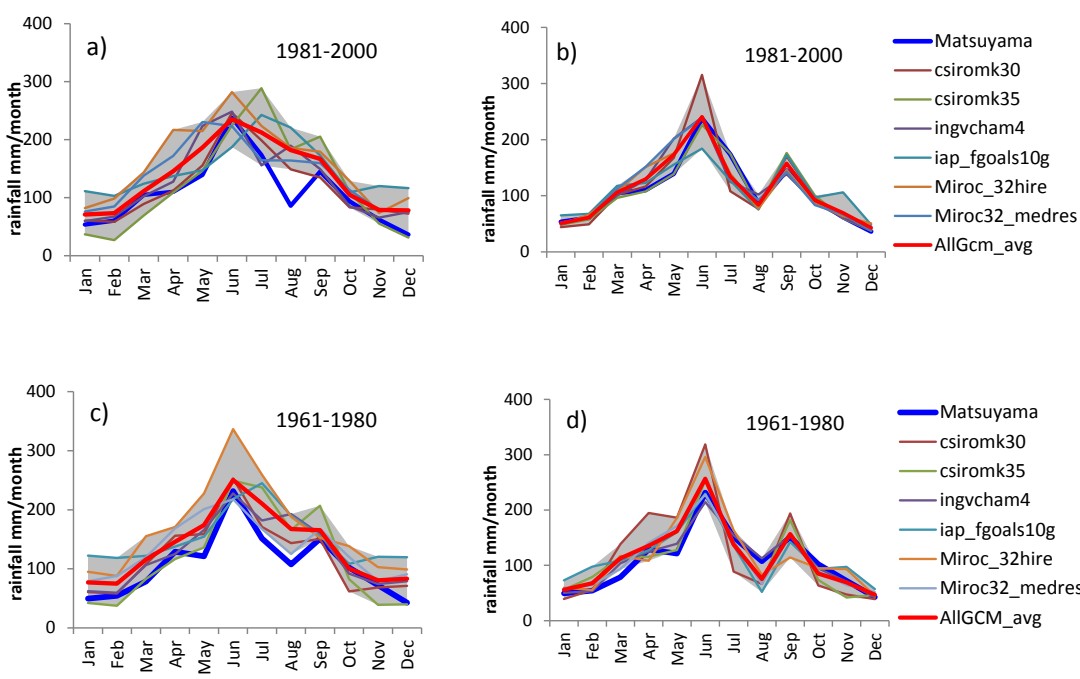

**Figure 11.** Seasonal climatology of Matsuyama station (Japan) (a, c) before bias correction and (b, d) after correction for calibration (1981–2000) and validation (1961–1980)





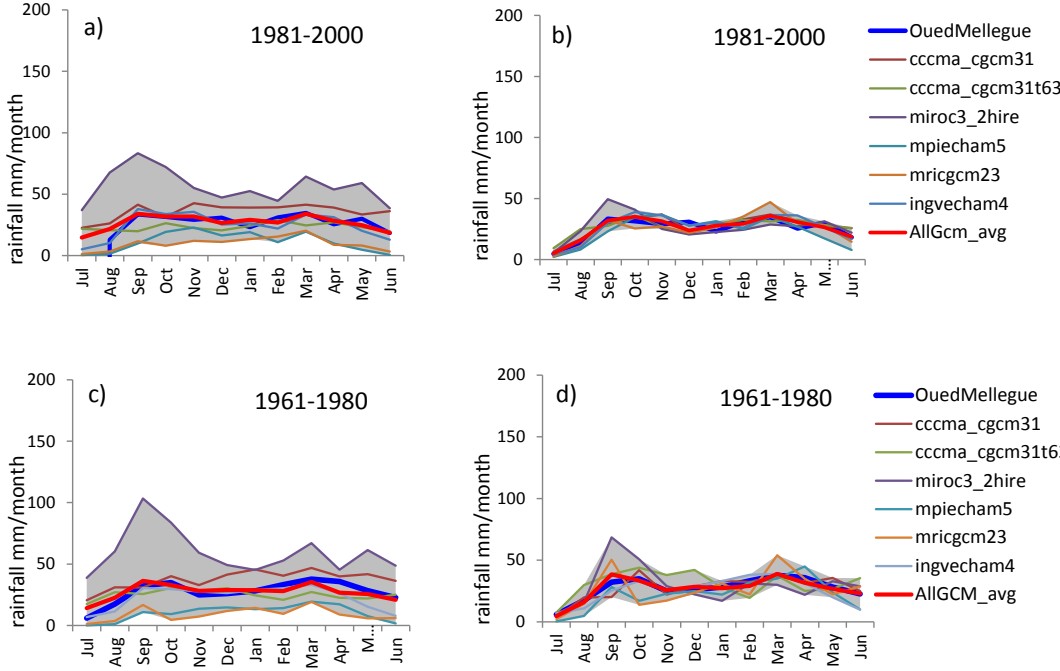

**Figure 12.** Seasonal climatology variation at Oued Mellegue station (Tunisia) (a, c) before bias correction and (b, d) after correction for calibration (1981–2000) and validation (1961–1980)

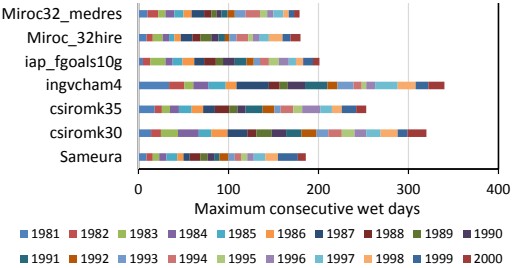

**Figure 13.** Twenty-year total maximum continuous wet day comparison with bias-corrected GCMs and Sameura station over 1981–2000



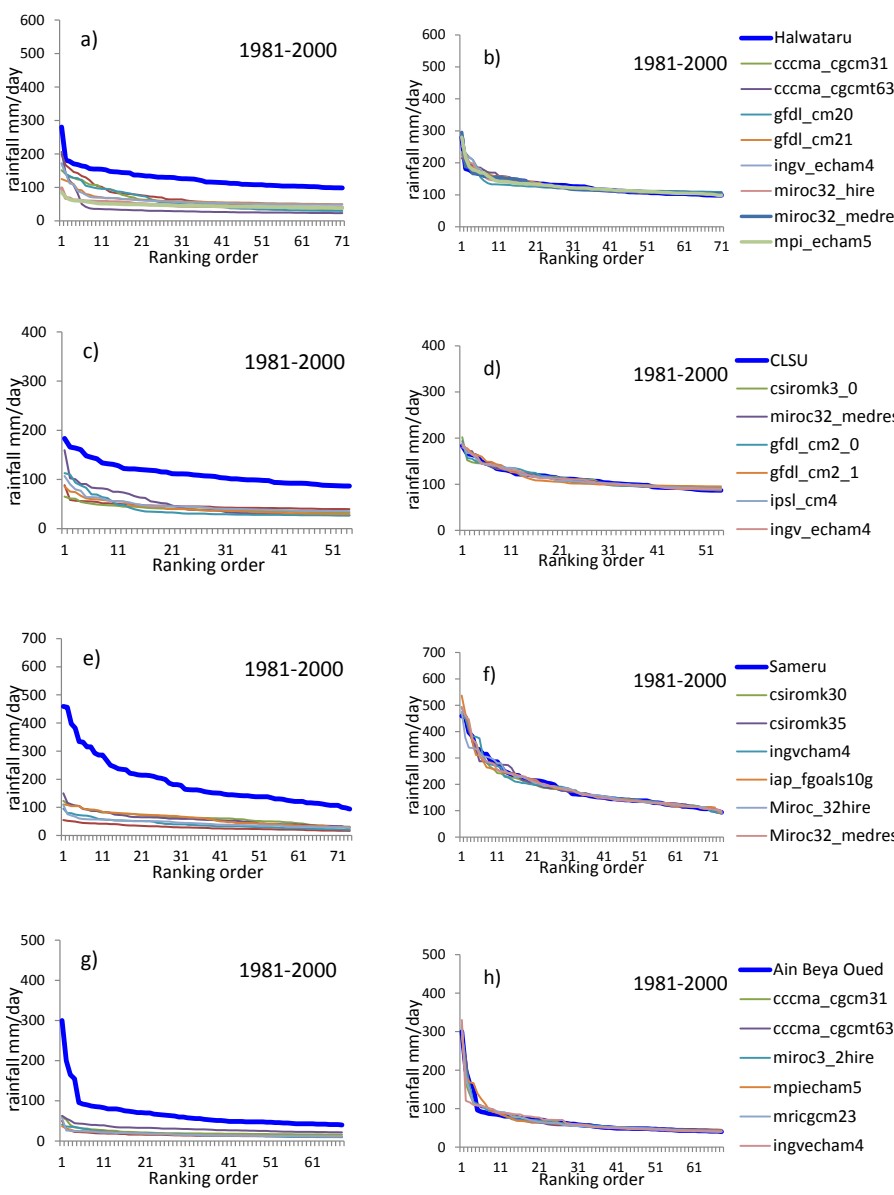

**Figure 14.** Bias-corrected extreme precipitation results (b, d, f and h) at Halwatru (Kalu Ganga), Central Luzon State University CLSU (Pampanga), Sameura (Yoshino), and Ain Beya Qued (Medjerda) stations, compared with raw GCM extreme precipitation (a, c, e and g)





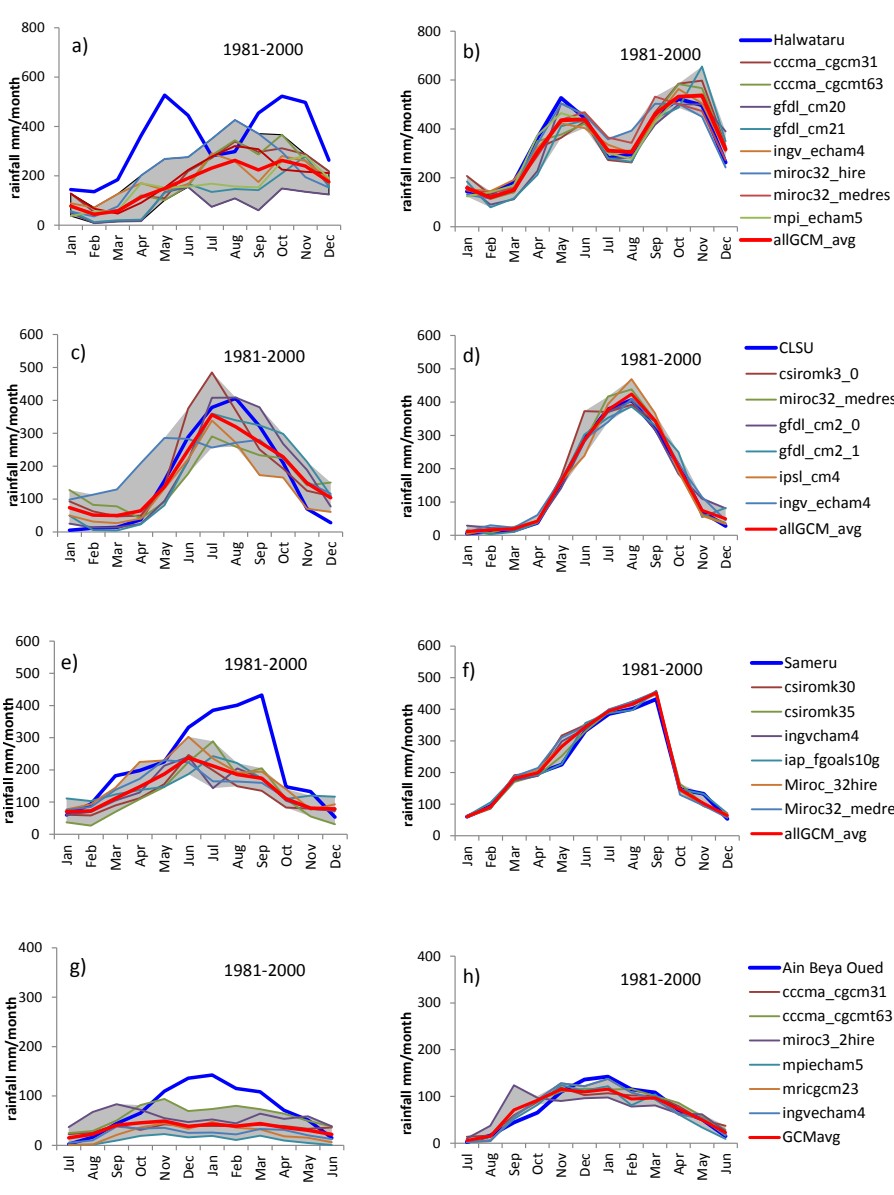

**Figure 15.** Bias-corrected monthly climatological precipitation (b, d, f and h) at Halwatru (Kalu Ganga), Central Luzon State University CLSU (Pampanga), Sameura (Yoshino) and Ain Beya Qued (Medjerda) stations, compared with before bias correction (a, c, e and g)




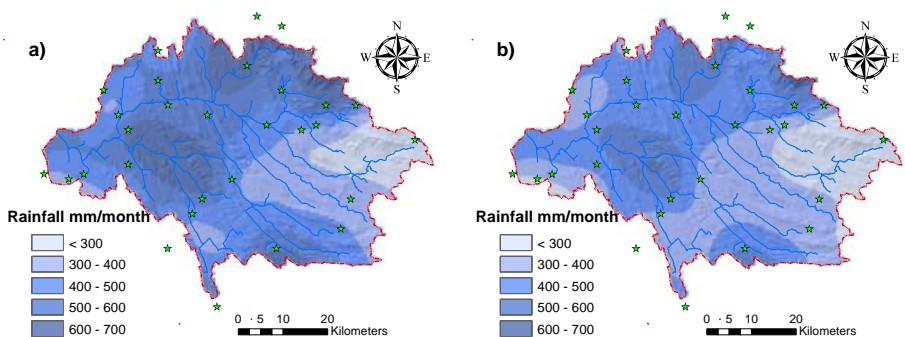

**Figure 16.** May climatological rainfall distribution at Kalu Ganga River: (a) observation; (b) bias-corrected GCM mean

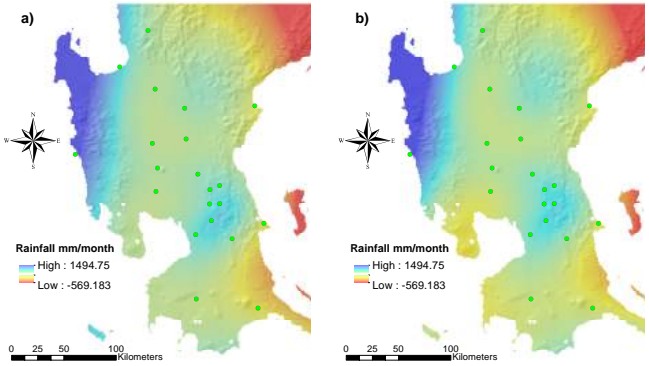

**Figure 17.** August climatological rainfall distribution for Pampanga, Angat and Kaliwa rivers: (a) observation; (b) bias-corrected GCM mean

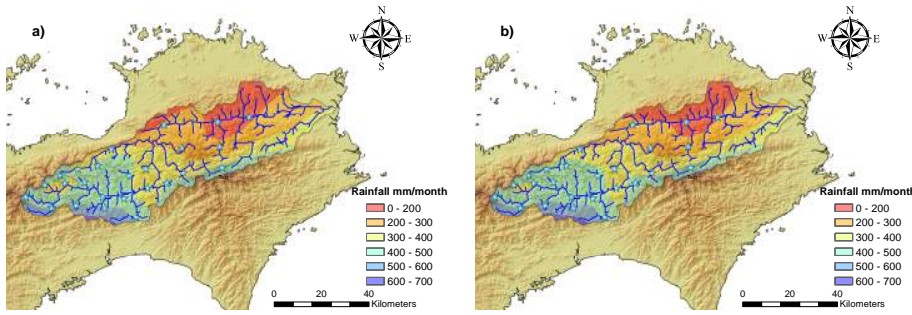

**Figure 18.** September climatology rainfall distribution in the Yoshino river: (a) observation; (b) bias corrected GCM mean





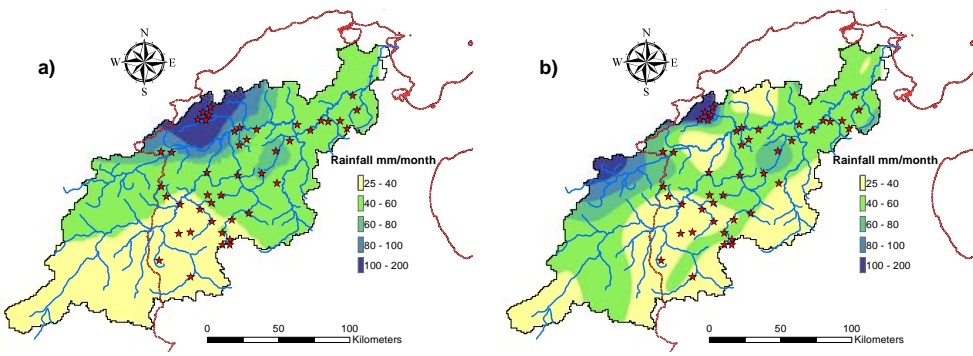

**Figure 19.** January climatological rainfall distribution for Medjreda river: (a) observation; (b) bias-corrected GCM mean

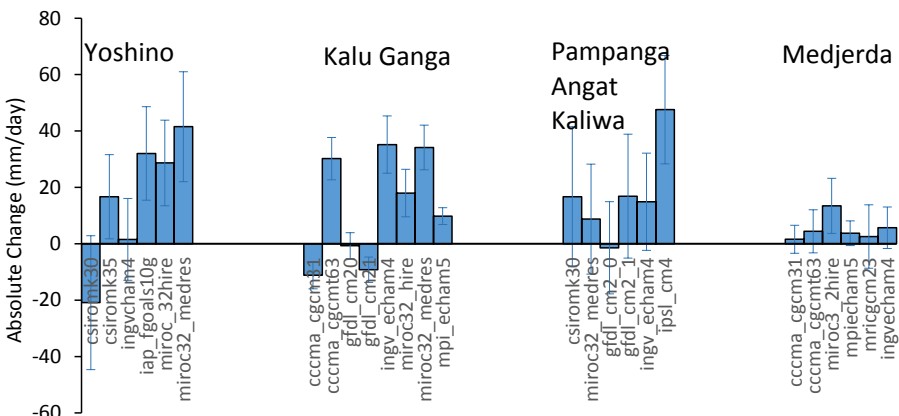

**Figure 20.** Absolute change of 99th percentile rainfall (based on ground stations' mean), using projected corrected GCM rainfall. Error bars indicate standard deviation among ground stations in each GCM for target basins

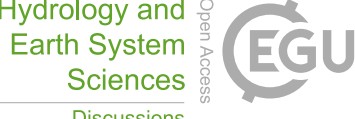

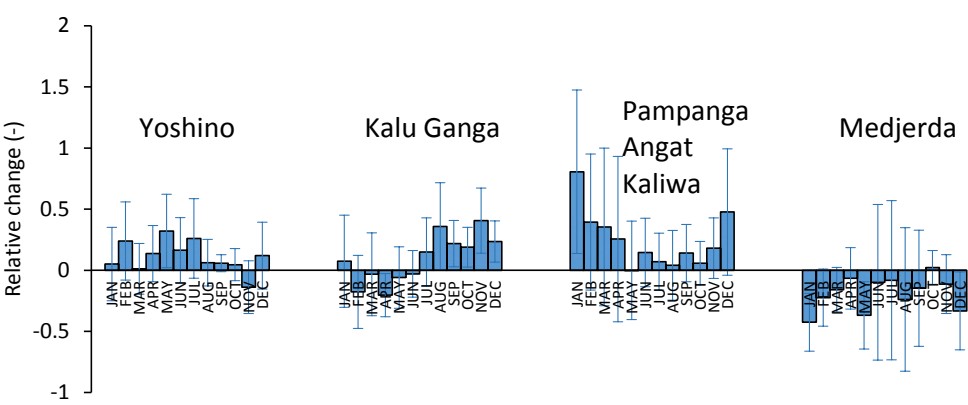

**Figure 21.** Projected relative change of climatological rainfall (based on observed), using mean of change in all GCMs calculated from average of stations in each GCM. Error bars indicate standard deviation among selected GCMs for target basins