# Peer review of "Statistical bias correction for climate change impact on the basin scale precipitation in Sri Lanka, Philippines, Japan and Tunisia"

_Hydrology and Earth System Sciences, 2016_

## Referee Comment (RC1) · Anonymous Referee #1 · 10 Mar 2016

Nyunt et al. introduced a new approach of correcting biases from GCM outputs, mainly precipitation, to study basin-scale climate change impact. While their 3-step bias correction approach accounting for biases in extreme events and wet-day frequency is interesting, there are a few scopes for improvement which should be addressed.

Specific comments:

1. The method of selecting GCM adopted in this study is not convincing. Addressing the uncertainties as reflected by the disagreements among the model simulations is a topic of extensive research. The authors tend to oversimplify this issue by selecting GCMs based on their "scores" used in this study. I understand that some criteria might be needed to narrow down the list of data sets to be analyzed. However, those criteria

should not appear as the main focus of this study. The way the approach of selecting GCMs was emphasized in abstract and in the first paragraph of introduction (from line 23 on page 1 to line 4 on page 2), readers might find it as one of the main objectives of this study.

2. The method of selecting GCMs prior to bias correction is, I think, contradictory to the core idea of this study. Since authors introduce a new approach to correct the model biases, to examine its efficiency, the best approach would be selecting a pool of GCMs with worse performance or larger uncertainties. Selecting only the better models somehow undermine the efficiency of the bias correction method.

3. Figs. 16-19 are useful, since spatial plots are often more telling in climate study. However, in addition to presenting the observation and bias-corrected model simulation, another plot representing the raw GCM mean precipitation before bias correction would be interesting. Also I am not sure why authors used different color scale for each of those spatial plots. A uniform color scale would be more informative. Figs. 16-19 can be combined into one single spatial plot where first, second and third column represent observation, raw GCM mean and bias-corrected GCM mean respectively, with each row representing a different river basin.

4. Since correcting biases of extreme precipitation is one of the major highlights of this study, a similar spatial plot comparing raw and bias-corrected GCM simulations with observed extreme precipitation over different river basins would be an interesting addition.

Technical Comment:

The introduction part should be more organized. Especially, in the third paragraph of introduction (line 19-35 on page 2), authors tend to go back and forth among three distinct points – 1) main focus/features of this study, 2) brief descriptions of methods, and 3) limitations of the approach. This seems to interrupt the flow of the manuscript.

---

## Short Comment (SC1) · 22 Mar 2016

I have reviewed with great interest the paper by Nyunt, et al. I have a short comment on the method the authors used to correct the future climate precipitation (equation 11 on page 8).

$$X'_{Gcm\_Fut} = F^{-1}_{Obs}(F_{Gcm\_past}(X_{Gcm\_Fut})) \text{———(11)}$$

According to the equation, the authors fed future precipitation ($X_{Gcm\_Fut}$) into the the CDF fitted using the past GCM precipitation data ($F_{Gcm\_Past}$) to get the cumulative probability. It means, they assumed that the future precipitation follows the same CDF as the past GCM precipitation. In practice, this is not true.

[Figure]

It is better to clarify why authors assumed the same CDF for the past and the future precipitation.

---

## Referee Comment (RC2) · Anonymous Referee #2 · 18 Apr 2016

SUMMARY In this study, a 3-step bias correction method is introduced to address, in a simultaneous way, three global climate model (GCM) deficiencies including underestimation of extremes, poor seasonal simulation, and high-frequency wet day error relative to observations. The introduced method depends on determining a regional pattern of climate variability through multi-model selection. The performance of the method is assessed for various climatic zones based on four catchments from Sri Lanka, Philippines, Japan, and Tunisia. One of the limitations of the proposed method is its reduced accuracy for low spatio-temporal scale of data.

The potential contribution of the study is interesting and relevant for climate change studies. However, because the authors need to first implement and/or address a number of issues to enhance the scientific quality of the contribution, I recommend major revision.

COMMENT No. 1 With respect to the text in lines 5-19 (page 2), this part of the Introduction Section is not adequately informative. For instance, what are the gaps in the existing bias correction methods such as the Delta method, distribution mapping, etc? How do the authors wish to close the identified gaps from other bias correction methods using the method they are proposing? The authors were somewhat jumpy instead of maintaining the required logical connections between the ideas. This made the Introduction Section not so well organized and needs an improvement.

COMMENT No. 2 In line 6 (page 3), each of the two periods 1981-2000 and 2046-2065 is 20 years in record length. However, according to the Intergovernmental Panel on Climate Change IPCC (2001), a 30-year period is sufficient and required to represent an effective GCM simulation. Can the authors justify that the projections and/or performance of the GCMs with respect to the meteorological variables considered based on the 20-year periods are not significantly different from that of the 30-year time frame recommended for climate change analyses?

COMMENT No. 3 In lines 2-3 (page 3), the authors stated they used the GCMs from previous generation i.e. phase 3 of the Coupled Model Inter-comparison Project (CMIP). The latest generation GCMs from phase 5 (CMIP5) are expectedly more improved than those of the CMIP3 especially with respect to bias in the extreme meteorological events. Recent climate change studies also seem to have tacitly adopted the use of rather the CMIP5 than CMIP3. In this same vein, can the authors clarify the rationale for the selection of the GCMs from rather the CMIP3 than CMIP5? On what basis did the authors select the few GCMs and their simulation runs?

COMMENT No. 4 Whereas the authors claim to have proposed an efficient bias correction approach for climate change impact investigation, in lines 30-31 (page 2), it is stated that the accuracy of the proposed method is contravened by spatio-temporal

scale of data.

i) Does it mean the proposed method cannot be usefully applied for data scarce regions? what options do the authors recommend to deal with this influence of data limitation on the accuracy of the proposed method?

ii) if validity of high resolution gridded freely available data (FAD) e.g. reanalysis or interpolated series can be verified, can the augmentation of the observed or historical datasets by such FAD enhance the applicability of the proposed method?

iii) In combination with the suggestion from (ii), could the downscaling procedure applied by Zhao et al. (2016) be useful to supplement bias correction procedures proposed by the authors?

Finally, I suggest that the authors acknowledge and also have the readers informed of the emphasis on the use of a number of bias correction techniques to even out the uncertainty due to the difference in the downscaling methods. In doing so, the authors may find the next three sentences constructive to include in their discussion as they acknowledge the limitation of their proposed method.

The differences between the downscaling methods and the performance of the bias correction approaches tend to vary from one catchment to another Sunyer et al. (2015). According to Sunyer et al. (2012) it is better to test the performance of different downscaling methods while importantly acknowledging their limitations, advantages as well as the downscaling uncertainties. Furthermore, statistical bias correction procedures should be applied on a case by case basis in line with the objectives of the climate change study (Onyutha et al., 2016), e.g. when dealing with moderate or extreme hydro-meteorological events.

COMMENT No. 5 Lines 17-18 (page 5): The main GCM deficiencies that the authors attempt to address include i) underestimation of extremes, ii) poor seasonal simulation, and iii) high-frequency wet day error relative to the observations. Although the

authors claim their proposed method attempts to solve the above GCM deficiencies in a simultaneous way not like by other bias correction techniques (as they state in lines 22-24 of page 2), I expect bias due to (i)-(iii) to be adequately addressed by the advanced quantile-perturbation-based (AQP) downscaling approach e.g. that presented by Willems and Vrac (2011). What advantages (if any) does the proposed method therefore have compared with, e.g. the AQP downscaling technique?

COMMENT No. 6 Lines 15-17 (page 6): "The heavy-tailed distributions are the most common in hydrology". Such naive generalizations are potentially misleading with respect to frequentist inference. According to Cai et al. (2013), the variables in meteorology and the environmental science generally exhibit the generalized Pareto distribution (GPD) shape parameter (k) around zero i.e. the normal tailed GPD. Even based on the data shown by the authors in Figure 6 (a) on page 23, it is noticeable that as the threshold becomes sufficiently large, indeed, the k tends towards zero. The necessitation of the estimators of the GPD parameter k to allow its estimation without having prior knowledge of its sign commonly tends to eliminate the need to make assumptions or fix the distribution class as heavy, normal or light-tailed (Onyutha and Willems, 2015). This generality leads to systematic bias (i.e. under/over-estimation) of quantiles in the tail of the GPD for some of the common parameter estimation methods such as the method of moments (which the authors applied), L-moment and maximum likelihood as demonstrated in Figure 8 of Onyutha and Willems (2015). Based on the above text and references, the authors are required to make it clear on the need to assess the class of the GPD before applying the proposed bias correction technique they are introducing. If possible, they should also ensure they incorporate the aspect of the discrimination between GPD classes as part of their proposed scheme for bias correction of extremes.

COMMENT No. 7 As proposed by Onyutha and Willems (2015), one way to minimize the bias in the quantiles from the tail of the GPD is to select the scale parameter ($\xi$) in an optimal way using graphical approach to identify the key event above which the

mean squared error on the GPD calibrated to the extreme events is minimal. The implementation of this proposal adequately by the authors based on Figure 6 a-b was a very good step in their proposed bias correction approach. However, some key parameter seems to be missing in equations (4) and (5).

It is well-known that: a) if the GPD parameter $\xi$ (threshold) is known, using the method of moment approach (as adopted by the authors), the shape (k) and scale ($\alpha$) parameters can be computed using:

k=0.5([($\mu$-$\xi$)/$\sigma$)]^2-1) ...............A1

$\alpha$=(1+k)($\mu$-$\xi$) .................A2

where $\mu$ and $\sigma$ denote the sample mean and standard deviation respectively.

b) if $\xi$ is unknown, method of moment estimates of k and $\alpha$ can be obtained using an iteration scheme (e.g. Newton-Raphson) from:

$\Psi$=2(1-k)(1+2k)^0.5/(1+3k) .......A3

$\xi$=$\mu$-$\alpha$/(1+k) ..................A4

k=$\sigma$(1+k)(1+2k)^0.5 ............A5

where $\Psi$ is the sample skewness.

Can the authors check the correctness of the their equations (4) and (5) on page 6 in comparison with those provided above i.e. A1 to A5?.

TECHNICAL CORRECTIONS (TC)

TC 1 Line 5 (page 1): delete "(Scorr)" since it was never used again within the abstract.

TC 2 Lines 15 (page 3): change "A total of ...............Sri Lanka" to "Rainfall data from a total of 26 stations were obtained from the Meteorology Department of Sri Lanka".

TC 3 Throughout the manuscript, the authors should be consistent with the use of

comma to separate thousands in figures (e.g. see, on page 3, lines 15, 19, 20, 22- 24, 28, etc).

TC 4 Instead of only mentioning the number of stations considered, the author should clearly specify the resolution of the data (e.g. daily, monthly etc) obtained from the different catchments. In the same vein, the data temporal domain for each hydro-meteorological variables used should also be included in Table 1.

TC 5 Line 4 (page4): replace "comparison to" with "in comparison with"

TC 6 Equation 2: is the RMSE the same as ERMS? Be consistent with one of the two for both text and equation(s).

TC 7 Line 28-29 (page4): What is i in xi or yi? "....and N the total time series". Do you want to mean N is the sample size of the series at each station? or is N the total number of time series? If N used in equation 2 is different in meaning from that of equation 7, the authors should not use the same notation.

TC 8 Lines 1-2 (page 5): For clarity, rephrase the sentence "By comparing all analysis months.................was implemented" to "To implement the scoring scheme, a particular GCM's average of the Scorr and RMSE obtained by considering the analyses from all the months were compared to the mean values of the 'goodness-of-fit' metrics (Scorr and RMSE) of all the GCMs."

TC 9 Line 6 (page 5): what is "ceratain"?

TC 10 Line 10 (page 5): "....we excluded GCMs that did not have a precipitation score of 1.....". It is possible that a particular GCM can have two or more simulation runs e.g. gfdl_cm2_1 and gfdl_cm2_0. I find it confusing whether the authors excluded the whole of such GCM or specifically the simulation runs with unsatisfactory scores. In case the entire GCM but not the simulation runs were excluded, which GCMs were those discarded? If the main purpose of the proposed method was bias correction, why should the GCMs with large bias be discarded? How can the authors verify the

efficacy of their proposed method if they ideally want to use the GCMs with minimum bias? The answers to these questions should be presented clearly.

TC 11 Lines 18 (page 5), 33 (page 2), 1 and 20 (page 3), 6 (page 4), 1 (page 1) ....etc, : "We did this and that...." such colloquial words do not have spaces for their accommodation in papers to be published by a top journal like HESS.

TC 12 Line 23 (page 5): replace "it just considers the top as extremes in a year basic, in other words" with "by considering the maximum event in each year, the"

TC 13 Line 24 (page 5): replace "year?s" with " year's"

TC 14 Line 25-26 (page 5): ".......20 maxima (1981-2000) by defining extremes that were larger than the smallest one." This sentence is unclear. Do you mean the Annual Maxima Series (AMS) extracted from 20-year (1981-2000) data? "... larger than the smallest one." which smallest one? Still if you refer to AMS, it is well-known that for the AMS extraction, the largest event in each hydro-meteorological year is selected.

TC 15 Line 27 (page 5): in using Partial Duration Series (PDS), the authors should clearly present the criteria they used to ensure that the key requirement of frequency analysis (viz the extreme events to be independent and identically distributed) was fulfilled.

TC 16 Line 28 (page 5): replace "of maxima" with " the extreme events"

TC 17 Line 28 (page 5): change "occurred" to " occur"

TC 18 Line 29 (page 5): insert "quantile" between "improve" and "estimation"

TC 19 Line 30 (page 5): change "rain" to "rainfall intensity"

TC 20 Lines 1-3 (page 6): delete the sentences "This is because all rain...........................bias correction efficiency". Replace "Therefore, the" with "The"

TC 21 Line 4 (page 6): replace "desirable" with "applied" and put a full stop after

"correction" not a comma

TC 22 Line 4 (page 6): replace "and" with "The GPD was used because"

TC 23 Line 5 (page 6): delete "variables using"

TC 24 Line 5 (page 6): replace "annual maximum flood" with "peak high flows"

TC 25 Equation 3 (page 6): define the symbol $\xi$ as it appears the first time (not as you did later in line 7 of page 7). What about x? is it similar to that used and defined in equations 1 and 2? The authors should make this clear.

TC 26 Line 17 (page 7): replace "tuned for the best fit" with "taken to be indicative of the best performance by the GCM".

TC 27 Line 24 (page 7): replace "We solved this problem" with "Attempt to apply bias correction for the frequency of wet days was made by".

TC 28 Line 26 (page 7): change "rain days" to "wet days". Implement this correction throughout the manuscript.

TC 29 Line 27 (page 7): Is the word "beyond" the same as "above"? If so, change it accordingly.

TC 30 Line 22 (page 8): replace "just get rid of" with "minimize"

TC 31 Line 9 (page 8): replace perfect"" with "reasonable"

TC 32 Line 29 (page 9): what is "?a?" ?

TC 33 Line 25 (page 11): what is basin?level?

TC 34 Figure 5 (page 22): these plots are neither so informative nor scientifically convincing with respect to the extreme value analysis which the authors claim to be considering. Since comparison of the quantiles from extremes extracted based on the AMS and PDS are being compared, why can't the quantile plots be made instead of showing the number of days and rainfall intensity? I recommend better plots than those in Figure 5 be made e.g. extreme rainfall intensity versus log-transformed return periods for better quantile-based assessment of bias. For an example of such plots, the authors can see Figure 3 of Sunyer et al. (2012), Figure 2 of Willems and Vrac (2011), Figure 4 of Onyutha et al. (2016), etc.

TC 35 Line 29 (page 13): replace "eliminates" with "reduces"

TC 36 The maps in Figures 1, and 16-19 should be presented with clearly marked grids and graticules to show locations (degrees of latitude and longitude) in geographic coordinates.

TC 37 For Figures 9 and 10 it cannot be understood that the letters a, b, ...and g in the horizontal axis represent the IDs of the GCMs as presented in Table 3 though stated in lines 28-29 of page 9. Make this clear in the Figure caption as well. No any difference seems noticeable by comparing the plot for the GCM 'a' before and after the application of the bias correction. How can the authors explain this realization? This should be clarified within the text of second paragraph in section of 3.4.

TC 38 Figure 14: compute the exceedance probability of each extreme rainfall event and use it to replace the ranking order plotted on the horizontal axis. I also recommend that throughout the manuscript, the expression "ranking order statistics" be replaced with "exceedance probability".

―――――――――――――――――

---

## Referee Comment (RC3) · Anonymous Referee #2 · 19 Apr 2016

References to be appended:

Cai, J.-J., deHaan,L. and Zhou, C.: Bias correction in extreme value statistics with index around zero. Extremes 16 (2), 173–201, 2013.

IPCC,: Climate Change: The Scientific Basis. Cambridge University Press, Cambridge and New York, NY, USA, p. 996, 2001.

Onyutha, P. , and Willems, P.: Uncertainty in calibrating generalised Pareto distribution to rainfall extremes in Lake Victoria basin. Hydrology Research, 46 (3) 356-376, 2015.

Onyutha, C., Tabari, H., Rutkowska, A., Nyeko-Ogiramoi, P., and Willems, P.: Comparison of different statistical downscaling methods for climate change rainfall projec-

tions over the Lake Victoria basin considering CMIP3 and CMIP5. Journal of Hydro-environment Research 12 (C), 31–45, 2016.

Sunyer, M.A., Madsen, H., and Ang P.H.: A comparison of different regional climate models and statistical downscaling methods for extreme rainfall estimation under climate change. Atmospheric Research 103, 119–128, 2012.

Sunyer, M. A., Hundecha, Y., Lawrence, D., Madsen, H., Willems, P., Martinkova, M., Vormoor, K., Bürger, G., Hanel, M., KriaučiÅńnienÄŮ, J., Loukas, A., Osuch, M., and Yücel, I.: Inter-comparison of statistical downscaling methods for projection of extreme precipitation in Europe. Hydrology and Earth System Sciences, 19, 1827-1847, 2015.

Zhao, N., Yue, T., Zhou, X., Zhao, M., Liu, Y., and Du, Z.: Statistical downscaling of precipitation using local regression and high accuracy surface modeling method. Theoretical and Applied Climatology. DOI: 10.1007/s00704-016-1776-z, 2016.

Willems, P., and Vrac, M.: Statistical precipitation downscaling for small-scale hydrological impact investigations of climate change. Journal of Hydrology, 402, 193–205, 2011.

---

## Author Comment (AC1) · 21 Jun 2016

**Statistical bias correction for climate change impact on the basin scale precipitation in Sri Lanka, Philippines, Japan and Tunisia**

*by* C.T. Nyunt et al.

Thanks for reviewing this manuscript and for your valuable comments which make our manuscript more constructive and informative. We mark our response as "AC: Author comment" and please see our responses as below:

Anonymous Refree#1
**Nyunt et al. introduced a new approach of correcting biases from GCM outputs, mainly precipitation, to study basin-scale climate change impact. While their 3-step bias correction approach accounting for biases in extreme events and wet-day frequency is interesting, there are a few scopes for improvement which should be addressed.**

**Specific comments:**

1. **The method of selecting GCM adopted in this study is not convincing. Addressing the uncertainties as reflected by the disagreements among the model simulations is a topic of extensive research. The authors tend to oversimplify this issue by selecting GCMs based on their "scores" used in this study. I understand that some criteria might be needed to narrow down the list of data sets to be analyzed. However, those criteria should not appear as the main focus of this study. The way the approach of selecting GCMs was emphasized in abstract and in the first paragraph of introduction (from line 23 on page 1 to line 4 on page 2), readers might find it as one of the main objectives of this study.**

**AC:** The method of selecting GCM adopted in this study is based on their performance scores of the synoptic scale simulation. It is a kind of standardized method or an adequate criterion to be generalized the way of selection (in line 2 from the abstract) under the conventional strategies of impact analysis. In other words, this method is the exclusion of improper GCMs which cannot express the significant regional climate characteristics in the reasonable spatial pattern and errors because the local seasonal precipitation is mainly governed by the synoptic scale features. Without prior poor GCMs rejection, the

considerable bias of GCM may be unfeasible and it is hard to believe the promising range of future quantitative outcomes. Hence, the proposed method convinces to disregard the undesirable GCMs.

The authors modified in "Abstract" and summarized (from line 1-6 on page 1) as follows: "A three-step statistical bias correction method is introduced to solve global climate model (GCM) bias after excluding the improper GCMs through the decisive scoring scheme. It is determined according to spatial correlation and root mean square error (RMSE) of regional and mesoscale climate variation in a comparison with global references."

In "Introduction", the paragraph (from line 23 on page 1 to line 4 on page 2) is deleted and modified as: "To ascertain feasible ranges of projected change, most studies recommend multi-GCM bias correction and simulations (Wilby et al., 2004; Paeth et al., 2008). Therefore, the present study introduces the three step statistical bias correction and the typical benchmark for discarding undesirable GCMs which cannot express the representative regional climate pattern to reduce specific contingencies in basin-scale impact assessment."

The title of Section 3.1 "Decrement of GCM uncertainty" is changed to "Exclusion of GCMs with poor regional climate characteristics".

**2. The method of selecting GCMs prior to bias correction is, I think, contradictory to the core idea of this study. Since authors introduce a new approach to correct the model biases, to examine its efficiency, the best approach would be selecting a pool of GCMs with worse performance or larger uncertainties. Selecting only the better models somehow undermine the efficiency of the bias correction method.**

AC: The proposed strategy is a standard way to access the climate change impact on the basin scale precipitation effectively and to save the heavy computation load. This kind of strategy is indispensable to provide the trustworthy uncertainty range for national and local infrastructures planning and IWRM. It is important to realize that GCMs simulation over the target basin showing a similar pattern with the feasible biases can be corrected by the proposed method and not for the GCMs with disparate pattern together with the enormous biases. Therefore, we proposed to discard some GCMs which are very poor in the region or / and the local scale performance. The selection method is a guide to exclude

the undesirable GCMs for impact assessment and not the way to choose the best GCMs (line 6-7 page 5). The multi GCM analysis can achieve the final goal to convince the reasonable range of impact analysis results for feasible implementation plan.

> **3. Figs. 16-19 are useful, since spatial plots are often more telling in climate study. However, in addition to presenting the observation and bias-corrected model simulation, another plot representing the raw GCM mean precipitation before bias correction would be interesting. Also I am not sure why authors used different color scale for each of those spatial plots. A uniform color scale would be more informative. Figs. 16-19 can be combined into one single spatial plot where first, second and third column represent observation, raw GCM mean and bias-corrected GCM mean respectively, with each row representing a different river basin.**

**AC:** Fig 16 - 19 are combined into one single plot as a Fig. 17 (Fig. 1 in the interactive comment) where first, second and third column represent observation, raw GCM mean and bias-corrected GCM mean respectively, with each row representing a different river basin. A uniform color scheme is used in the first, second and third row of spatial plots but semi-arid basin from Tunisia, the last one, cannot be in the same scale because of its low intensity per month.

> **4. Since correcting biases of extreme precipitation is one of the major highlights of this study, a similar spatial plot comparing raw and bias-corrected GCM simulations with observed extreme precipitation over different river basins would be an interesting addition.**

**AC:** The authors really appreciate the valuable comments about correcting biases of extreme precipitation and prepared Fig. 15 (Fig. 2 in the interactive comment) as a similar spatial distribution plot for 95 percentile precipitation of total rain days in all of the river basins. Similarly, first, second and third column represents observation, raw GCM mean and bias-corrected GCM mean respectively. Same color scheme for top three rows of figures with different maximum limit and smaller scale for the last basin in Tunisia.
The explanation of Figure 15 is added in Section 4.2 from line 26 on page 11 to line 2 on page 12 as below:
"To confirm the basin-scale performance, we choose 95 percentile rainfall (95Ex) from the total wet days during the control period at every point in the pilot basins. For the

spatial pattern checking, the spline interpolation is used to delineate in each basin. The summary of all river basins is shown in Fig.15 and panel a, b and c represent observation, raw GCM mean and bias corrected GCM mean respectively. In general, all of raw GCM mean 95Ex (panel (b)) undervalue compared to the observation (panel (a)). Distributions of bias corrected rainfall (panel (c)) from each river basin are noticeable matching with ground data except the top row where corrected one shows slightly low intensity in the middle of a sparse rain gauges. This difference occurs due to the balance of GPD fitting between the most extreme and lower intensity rainfall. As a result, imperfect fitting may happen in somewhere of the basin. Moreover, bias corrected results show around $\pm$ 5~20 mm day$^{-1}$ compared to the ground and the performance looks good enough for outlier rainfall bias-correction in all of the basins. In fact, it is important to realize both the station and basin level extreme bias-correction is successfully worked out."

**Technical Comment:**

**The introduction part should be more organized. Especially, in the third paragraph of introduction (line 19-35 on page 2), authors tend to go back and forth among three distinct points – 1) main focus/features of this study, 2) brief descriptions of methods, and 3) limitations of the approach. This seems to interrupt the flow of the manuscript.**

**AC:** Authors reorganize "Introduction" as the following. First paragraph explains why multi-GCM simulation is necessary for local scale studies, the procedures and goal of the proposed method. Second one briefly outlines the various bias correction techniques and the last one summarize the scope, how to treat GCM selection process, the specialty, strategy, successes of the proposed method, an assumption theory, a limitation and then a summary.

The third paragraph from "Introduction" is modified as below:

"The scope of the present study is to access the impact of climate change on the basin scale precipitation through multi-GCM bias correction. Defective GCMs are judged by the convectional scoring scheme based on the pattern and error analysis of the distinguished features of synoptic scale seasonal characteristics. The major outstanding factor of the proposed bias correction method is its simultaneous action in addressing all GCM deficiencies, such as underestimation of extremes, poor simulation of inter-seasonal rain sequences, and wet day errors whereas most correction approaches treat only either one of them. The major achievement is excluding biases in both amplitudes and

frequencies of GCM precipitation time series on an inter-annual and yearly basis, which contributes to the projection of basin-scale flood and drought analysis. Furthermore, the transferability of the application is confirmed by its implementation in three river basins in Asia and one in the Mediterranean region under diverse climates and local spatial variation as the benchmark of choice. The assumption of time-invariant GCM bias during control and projection periods is verified by two decadal simulations, 1981–2000 as the calibration period and 1961–1980 as the evaluation period, at a few ground locations where long-term data are available. A limitation of the approach is that reasonable numbers of well-scattered stations with a few decades of data are prerequisite. In this way, a comprehensive and integrated statistical bias correction and spatial downscaling method together with tackling poor GCMs concern was developed. Particularly, analysis of potential impacts on basin-scale precipitation is successfully achieved with low computation and high efficacy."

**References**

Wilby, R., Charles, S., Zorita, E., Timbal, B., Whetton, P., and Mearns, L.: Guidelines for use of climate scenarios developed from statistical downscaling methods, IPCC task group on data and scenario support for impacts and climate analysis, 2004.

Paeth, H., Scholten, A., Friederichs, P., and Hense, A.: Uncertainties in climate change prediction: El Niño-Southern Oscillation and monsoons, Global and Planetary Change, 60, 265–288, 2008.

---

## Author Comment (AC2) · 21 Jun 2016

**Statistical bias correction for climate change impact on the basin scale precipitation in Sri Lanka, Philippines, Japan and Tunisia**

*by* C.T. Nyunt et al.

Thanks for reviewing and your short comments which make our manuscript more constructive and informative. We mark our response as "AC: Author comment" and please see our responses as below:

**Short Comment from P. Hunukumbura**
**I have reviewed with great interest the paper by Nyunt, et al. I have a short comment on the method the authors used to correct the future climate precipitation (equation 11 on page 8).**

$$X'_{Gcm\_Fut} = F^{-1}_{Obs}\left(F_{Gcm\_past}\left(X_{Gcm\_Fut}\right)\right) \qquad (11)$$

**According to the equation, the authors fed future precipitation ($X_{Gcm\_Fut}$) into the the CDF fitted using the past GCM precipitation data ($F_{Gcm\_Past}$) to get the cumulative probability. It means, they assumed that the future precipitation follows the same CDF as the past GCM precipitation. In practice, this is not true. It is better to clarify why authors assumed the same CDF for the past and the future precipitation.**

**AC:** $x'_{GCM\_Fut} = F^{-1}_{Obs\&GCM\_past}\left(x_{GCM\_Fut}\right)$

Equation 11 is modified as above and the main theme is the bias discarded during the control period remain the same in the future period. As a consequence, the same inverse function between the observed CDF and original GCM CDF is used for bias correction of GCMs future precipitation. In other words, future is unknown and the stationary bias assumption (Section 3.4) has been confirmed by using the same function to the different 20 years (1961-1980) simulation at the two stations as the validation where the continuous 40 years (1961-2000) data available. For this reason, we don't assume future GCM precipitation follows the same CDF as the past and we just use the same inverse function between observation and GCMs during the control period for projected precipitation.

---

## Author Comment (AC3) · 21 Jun 2016

**Statistical bias correction for climate change impact on the basin scale precipitation in Sri Lanka, Philippines, Japan and Tunisia**

*by* C.T. Nyunt et al.

Thanks for reviewing this manuscript and for your valuable comments which make our manuscript more constructive and informative. We mark our response as "AC: Author comment" and please see our responses as below:

Anonymous Referee # 2
**SUMMARY In this study, a 3-step bias correction method is introduced to address, in a simultaneous way, three global climate model (GCM) deficiencies including underestimation of extremes, poor seasonal simulation, and high-frequency wet day error relative to observations. The introduced method depends on determining a regional pattern of climate variability through multi-model selection. The performance of the method is assessed for various climatic zones based on four catchments from Sri Lanka, Philippines, Japan, and Tunisia. One of the limitations of the proposed method is its reduced accuracy for low spatio-temporal scale of data. The potential contribution of the study is interesting and relevant for climate change studies. However, because the authors need to first implement and/or address a number of issues to enhance the scientific quality of the contribution, I recommend major revision.**

**COMMENT No. 1 With respect to the text in lines 5-19 (page 2), this part of the Introduction Section is not adequately informative. For instance, what are the gaps in the existing bias correction methods such as the Delta method, distribution mapping, etc? How do the authors wish to close the identified gaps from other bias correction methods using the method they are proposing? The authors were somewhat jumpy instead of maintaining the required logical connections between the ideas. This made the Introduction Section not so well organized and needs an improvement.**

**AC:** The authors modified according to the comments about line 5-19 (page 2) as the following in the Introduction.
"A variety of statistical bias correction methods have been developed and there are three main groups such as simple linear factor or delta change or multiplicative correction

method (Hay et al., 2000; Graham et al., 2007), power laws or non-linear correction (Bordoy and Burlando, 2013; Leander and Buishand, 2007; Hurkmans et al., 2010) and distribution mapping also known as quantile mapping or histogram equalization or matching (Wood et al., 2004; Schoetter et al., 2012; Piani and Haerter, 2012; Maurer et al., 2013). The simple linear method just treats the monthly mean intensity error and does not account for the difference in the frequency distribution of precipitation while non-linear correction of either twelve months or some days is possible to adjust both the mean and the spread or the coefficient of variation of rainfall distribution. Moreover, it removes the bias of intermediate rainfall very well, but still extreme corrected rainfall shows underestimation. These aforementioned methods do not cover the wet day frequency errors. In the distribution based correction method by the empirical or gamma, the bias of the mean and standard deviation together with wet-day frequencies and intensity errors are corrected simultaneously in monthly scale. This approach has been widespread use and show superior performance to other methods, because high moment biases, including wet day frequencies are effectively discarded (Ines and Hansen, 2006; Li et al., 2010; Piani et al., 2010a, b; Argüeso et al., 2013; Lafon et al., 2013; Teutschbein and Seibert, 2013).

Because it is more straightforward and incurs less computational burden, this approach has become a popular and conventional tool for GCM bias correction. However, the performance depends on how well do the observation and GCM precipitation fit the specific distribution (e.g. Extreme rainfall does not follow the normal distribution fitting). For this reason, the proposed method mixes two probability functions, one for tailed intensities that usually do not conform to a normal distribution and the other for a monthly series of normal precipitation, including adjustment of wet day frequency errors. As it is a statistic based correction method, it has no temporal correlation of rainfall series and the mismatch of temporal evolution may alter the further impact studies of the hydrological simulation. Moreover, the differences between the downscaling methods and the performance of the bias correction approaches tend to vary from one catchment to another (Sunyer Pinya et al., 2015). According to Sunyer et al. (2012) it is better to test the performance of different downscaling methods while importantly acknowledging their limitations, advantages as well as the downscaling uncertainties. Furthermore, statistical bias correction procedures should be applied on a case by case basis in line with the objectives of the climate change study (Onyutha et al., 2016), e.g. when dealing with moderate or extreme hydro-meteorological events."

**COMMENT No. 2 In line 6 (page 3), each of the two periods 1981-2000 and 2046-2065 is 20 years in record length. However, according to the Intergovernmental Panel on Climate Change IPCC (2001), a 30-year period is sufficient and required to represent an effective GCM simulation. Can the authors justify that the projections and/or performance of the GCMs with respect to the meteorological variables considered based on the 20-year periods are not significantly different from that of the 30-year time frame recommended for climate change analyses?**

**AC**: In the Intergovernmental Panel on Climate Change IPCC (2001), climate is defined in glossary as "Climate in a narrow sense is usually defined as the average weather, or more rigorously, as the statistical description in terms of the mean and variability of relevant quantities over a period of time ranging from months to thousands or millions of years. The classical period for averaging these variables is 30 years, as defined by the World Meteorological Organization. The relevant quantities are most often surface variables such as temperature, precipitation and wind. Climate in a wider sense is the state, including a statistical description, of the climate system. In various chapters in this report different averaging periods, such **as a period of 20 years**, are also used."
http://www.ipcc.ch/publications_and_data/ar4/wg1/en/annex1sglossary-a-d.html

Accordingly, the two periods 1981-2000 and 2046-2065 (20 years) simulation results are also in the tolerable range for the effective GCMs simulations although it is not the classical period (30-years) for climate change analysis. In this study, most of the ground station data from different river basins are available from 1981 to 2000.

In Matsuyama station in Japan, daily precipitation from 1961 to 2000 is available. The multi GCMs mean show the climatology value of 250.41 mm in 20-year (1961-1980) average and 241.76 mm in 30-year average (1961-1990) during the highest rainfall month of June in the following figure. This figure compares the 20-year seasonal climatology mean with 30-year seasonal climatology mean. As in the following figure, 1961-1980 (20 years) climatology variation is not significantly different from 1961-1990 (30years) climate statistics. Therefore, the performance of GCMs based on the 20-year periods are not significantly different from 30-year analysis.

[Figure]

**COMMENT No. 3 In lines 2-3 (page 3), the authors stated they used the GCMs from previous generation i.e. phase 3 of the Coupled Model Inter-comparison Project (CMIP). The latest generation GCMs from phase 5 (CMIP5) are expectedly more improved than those of the CMIP3 especially with respect to bias in the extreme meteorological events. Recent climate change studies also seem to have tacitly adopted the use of rather the CMIP5 than CMIP3. In this same vein, can the authors clarify the rationale for the selection of the GCMs from rather the CMIP3 than CMIP5? On what basis did the authors select the few GCMs and their simulation runs?**

**AC:** The main reason for the selection of the GCMs from rather the CMIP3 than CMIP5 is that all full sets of GCM data from CMIP3 have been archived on the DIAS server and CMIP5 GCMs data archive is ongoing due to the very huge amount data. Full sets of CMIP5 GCMs data are not completely uploaded on the DIAS server. This paper is focused on the development of the comprehensive bias correction method by ground observation and also the criteria of the exclusion of GCMs which cannot express the representative regional climate pattern over the target basin. Although most of CMIP5 GCMs are improved their performances, they still need further advancement to solve some issues. Therefore, the improved CMIP5 GCMs also need to evaluate under the same basic principles for omission of GCMs. But an extra remark for GCMs and their simulation runs is the GCM ensembles mean of different simulation runs are considered for the selection from CMIP5. After that, each run is treated as each simulation in the statistical bias correction.

**COMMENT No. 4 Whereas the authors claim to have proposed an efficient bias correction approach for climate change impact investigation, in lines 30-31 (page 2),**

**it is stated that the accuracy of the proposed method is contravened by spatio-temporal scale of data.**

> **i)** **Does it mean the proposed method cannot be usefully applied for data scarce regions? what options do the authors recommend to deal with this influence of data limitation on the accuracy of the proposed method?**

**AC:** If the ground station network does not represent the basin area average precipitation very well, in other way, spatial distribution of observed network is very poor, the proposed method can be implemented using the freely available data like GSMaP (Global Satellite Mapping of Precipitation) which is a high-precision, high resolution global precipitation map using satellite data from 1998 to until now. The data is $0.25° \times 0.25°$ and $0.1° \times 0.1°$ rain map with daily or hourly or monthly scale. But there is some bias in GSMaP rain map compared to the ground observation and the rain gauge adjusted GSMap rain map may be useful for the basin with sparse stations.

http://sharaku.eorc.jaxa.jp/GSMaP_crest/index.html

Another one is APHRODITE (Asian Precipitation - Highly-Resolved Observational Data Integration Towards Evaluation) daily gridded precipitation. It contains a dense observation network of daily rain gauge data for Asia including the Himalayas, South and Southeast Asia and mountainous areas in the Middle East since 1951. Because of its high density and quality station network with $0.5°$ or $0.25°$ grid resolution, it is a kind of surrogate surface precipitation but it is limited only for Asia. [Retrieved from https://climatedataguide.ucar.edu/climate-data/aphrodite-asian-precipitation-highly-resolved-observational-data-integration-towards.]

Another substitute is ERA-Interim data which is the third generation reanalysis and available from 1979 to present with $0.75° \times 0.75°$ spatial resolution with sub-daily and daily scale precipitation.

[https://climatedataguide.ucar.edu/climate-data/era-interim.]

However, these data cannot be used where the catchment area is smaller than a grid resolution.

> **ii)** **if validity of high resolution gridded freely available data (FAD) e.g. reanalysis or interpolated series can be verified, can the augmentation of the observed or historical datasets by such FAD enhance the applicability of the proposed method?**

**AC:** In Rasmy et. al (2013), the proposed method (statistical bias correction) is applied to the Dynamical Downscaled (DD) rainfall which is resulted from the Weather Research and Forecasting (WRF) by using the initial and boundary conditions from the ERA-interim data over Shikoku Island, Japan for the present climate. Owing to the direct influence of the spatial resolution enhancement in WRF, it was able to reproduce similar pattern and statistics as the observed rainfall.  However, there is still bias of overestimation, underestimation and frequency of wet day errors. But it has a much smaller bias than GCM rain and these biases are eliminated by statistical approach. Finally, future changes will be assessed using Pseudo Global Warming Downscale (PGW-DS) using the monthly mean difference between the future and present climates simulated by GCMs (Kawase et al., 2009). Therefore, it can be concluded that the proposed method enhances the high resolution gridded FAD or interpolated series or reanalysis rainfall for impact assessment studies and recognizes its applicability to global FAD.

iii)   **In combination with the suggestion from (ii), could the downscaling procedure applied by Zhao et al. (2016) be useful to supplement bias correction procedures proposed by the authors?**

**AC:** Both Dynamical (as in (ii)) and Statistical downscaling procedures are useful to implement the high spatial resolution precipitation surface which is very important for the basin with high intensity variation and further hydrological impact on river flow. As in Zhao et al. (2016), the weight based on geography (GWR) model based regression function and using station data to modify the residuals by high accuracy surface modelling method (HASM) are very good approach. But their results focus on the annual and seasonal scale downscaled results. It may be more interesting if the downscaled results are shown in the higher temporal resolution (e.g. monthly or daily) than annual and seasonal scale. For IWRM planning, not only annual and seasonal but also monthly or daily is important especially for the extreme events.

**Finally, I suggest that the authors acknowledge and also have the readers informed of the emphasis on the use of a number of bias correction techniques to even out the uncertainty due to the difference in the downscaling methods. In doing so, the authors may find the next three sentences constructive to include in their discussion as they acknowledge the limitation of their proposed method.**
**The differences between the downscaling methods and the performance of the bias correction approaches tend to vary from one catchment to another Sunyer et al.**

**(2015). According to Sunyer et al. (2012) it is better to test the performance of different downscaling methods while importantly acknowledging their limitations, advantages as well as the downscaling uncertainties. Furthermore, statistical bias correction procedures should be applied on a case by case basis in line with the objectives of the climate change study (Onyutha et al., 2016), e.g. when dealing with moderate or extreme hydro-meteorological events.**

**AC:** The three sentences constructive are included in "Introduction".

**COMMENT No. 5 Lines 17-18 (page 5): The main GCM deficiencies that the authors attempt to address include i) underestimation of extremes, ii) poor seasonal simulation, and iii) high-frequency wet day error relative to the observations. Although the authors claim their proposed method attempts to solve the above GCM deficiencies in a simultaneous way not like by other bias correction techniques (as they state in lines 22-24 of page 2), I expect bias due to (i)-(iii) to be adequately addressed by the advanced quantile-perturbation-based (AQP) downscaling approach e.g. that presented by Willems and Vrac (2011). What advantages (if any) does the proposed method therefore have compared with, e.g. the AQP downscaling technique?**

**AC:** AQP temporal downscaling approach modified historical series of ground data (Willems and Vrac, 2011) by using the quantiles based perturbation factor to reproduce the high temporal resolution of the projected rainfall for future urban planning. The effect of GCM bias of precipitation was not taken into account. It just adds the delta change between the original and scenario GCM to the historical observed data for analysis of scenario impacts. In other word, the span of the original amplitudes between different climate models for impact change is modified by imposing the future changes on the historical observed data without considering the systematic bias of multi-GCMs. Therefore, it may be some doubt in the final decision of the future urban design storm.

However, the range come from bias corrected GCMs (the proposed method) looks narrow down for easy decision as in Fig. 16 (b, d, f, h) (as Fig. 1 in the interactive comment). Moreover, the low exceedance probability rainfall (i.e., extremes) over a certain threshold are distinguished from all analysis years (not the monthly scale as in Willems and Vrac, 2011) to construct the heavy tailed representative distribution mapping for bias correction. Therefore, bias corrected function based on the transfer of CDF between observed and historical GCMs means a key to discard some systematic error of

GCMs biases. Hence, the maximum probable rainfall estimates from extremes in different return period for future plans may be more informative than monthly based analysis. On the other hand, the effectiveness of distribution mapping bias correction may depend on how the data are well-fitted to the assumed distribution.

**COMMENT No. 6 Lines 15-17 (page 6): "The heavy-tailed distributions are the most common in hydrology". Such naive generalizations are potentially misleading with respect to frequentist inference. According to Cai et al. (2013), the variables in meteorology and the environmental science generally exhibit the generalized Pareto distribution (GPD) shape parameter (k) around zero i.e. the normal tailed GPD. Even based on the data shown by the authors in Figure 6 (a) on page 23, it is noticeable that as the threshold becomes sufficiently large, indeed, the k tends towards zero. The necessitation of the estimators of the GPD parameter k to allow its estimation without having prior knowledge of its sign commonly tends to eliminate the need to make assumptions or fix the distribution class as heavy, normal or light-tailed (Onyutha and Willems, 2015). This generality leads to systematic bias (i.e. under/over-estimation) of quantiles in the tail of the GPD for some of the common parameter estimation methods such as the method of moments (which the authors applied), L-moment and maximum likelihood as demonstrated in Figure 8 of Onyutha and Willems (2015). Based on the above text and references, the authors are required to make it clear on the need to assess the class of the GPD before applying the proposed bias correction technique they are introducing. If possible, they should also ensure they incorporate the aspect of the discrimination between GPD classes as part of their proposed scheme for bias correction of extremes.**

**AC:** We modified the line 14-18 (page 6) as the following;

"Hosking and Wallis (1987) proposed different methods such as the maximum likelihood (ML), methods of moments (MOM) and probability weighted moments (PWM) estimators for estimating the parameters of the GPD in the case $\kappa > -0.5$ and PWM and MOM are extremely simple to compute. The PWM method may be appropriate for $\kappa < -0.2$ which means GPD with an extremely long tail and the ML method should be used in the case of $\kappa > 0.2$ with large sample sizes to avoid a high rate of inconsistency with the data. For the small to moderate sample sizes, the MOM or the PWM performed better than ML estimators (Tajvidi, 2003). The MOM is suggested to use when the samples are neither an extremely long tail nor an extremely short tail (de Zea Bermudez and Turkman, 2003) and it is the most efficient method for negative shape parameters of

the GPD model for estimation of quantile of T-year return period (Madsen et al., 1997). Therefore, it seems relevant to use the shape parameter κ in the range of -0.5 < κ < 0.5 (Rosbjerg et al., 1992) for any practical application. This limitation is same as in this study and the shape and scale parameters given by the MOM method are;"

**COMMENT No. 7 As proposed by Onyutha and Willems (2015), one way to minimize the bias in the quantiles from the tail of the GPD is to select the scale parameter ($\xi$) in an optimal way using graphical approach to identify the key event above which the mean squared error on the GPD calibrated to the extreme events is minimal. The implementation of this proposal adequately by the authors based on Figure 6 a-b was a very good step in their proposed bias correction approach. However, some key parameter seems to be missing in equations (4) and (5).**
**It is well-known that: a) if the GPD parameter $\xi$ (threshold) is known, using the method of moment approach (as adopted by the authors), the shape ($k$) and scale ($\alpha$) parameters can be computed using:**

$$k = 0.5\left(\left[(\mu - \xi)/\omega\right]^2 - 1\right) \text{ --------A1}$$

$$\alpha = (1 + k)(\mu - \xi) \text{------------ A2}$$

**where $\mu$ and $\sigma$ denote the sample mean and standard deviation respectively.**
**b) if $\xi$ is unknown, method of moment estimates of $k$ and $\alpha$ can be obtained using an iteration scheme (e.g. Newton-Raphson) from:**

$$\psi = 2(1 - k)(1 - 2k)^{0.5}/(1 + 3k) \text{ ------------------A3}$$

$$\xi = \mu - \alpha/(1 + k) \text{ ------------------ A4}$$

$$k = \sigma(1 + k)(1 + 2k)^{0.5} \text{------------------ A5}$$

**where $\psi$ is the sample skewness.**
**Can the authors check the correctness of their equations (4) and (5) on page 6 in comparison with those provided above i.e. A1 to A5?.**

**AC:** The equations are valid for the GPD parameter ξ (threshold) is known and in the case of the shape parameter κ in the range of -0.5 < κ < 0.5 as in Hosking and Wallis (1987) and Martins and Stedinger (2001).

**TECHNICAL CORRECTIONS (TC)**

**TC 1 Line 5 (page 1): delete "(Scorr)" since it was never used again within the abstract.**

**AC:** Deleted "(Scorr)"

**TC 2 Lines 15 (page 3): change "A total of ...............Sri Lanka" to "Rainfall data from a total of 26 stations were obtained from the Meteorology Department of Sri Lanka".**

**AC:** Changed.

**TC 3 Throughout the manuscript, the authors should be consistent with the use of comma to separate thousands in figures (e.g. see, on page 3, lines 15, 19, 20, 22- 24, 28, etc).**

**AC:** Added comma to separate thousands on page 3.

**TC 4 Instead of only mentioning the number of stations considered, the author should clearly specify the resolution of the data (e.g. daily, monthly etc) obtained from the different catchments. In the same vein, the data temporal domain for each hydro-meteorological variables used should also be included in Table 1.**

**AC:** Added "Daily" in Line15, 20, 25, 29 on page 3. Analysis period and time scale columns are added in Table 1.

**TC 5 Line 4 (page4): replace "comparison to" with "in comparison with"**

**AC:** Replaced.

**TC 6 Equation 2: is the RMSE the same as ERMS? Be consistent with one of the two for both text and equation(s).**

**AC:** $RMSE = \sqrt{\dfrac{1}{n}\sum_{i=1}^{n}\left(p_i - q_i\right)^2}$

**TC 7 Line 28-29 (page4): What is i in xi or yi? "....and N the total time series". Do you want to mean N is the sample size of the series at each station? or is N the total number of time series? If N used in equation 2 is different in meaning from that of equation 7, the authors should not use the same notation.**

**AC:** Changed to "n" and i in $x_i$ or $y_i$ mean a specific month, n is the total time series.

$$S_{corr} = \frac{\sum_{i=1}^{n}\left(p_i - \overline{p}\right)\left(q_i - \overline{q}\right)}{\sqrt{\sum_{i=1}^{n}\left(p_i - \overline{p}\right)^2\left(q_i - \overline{q}\right)^2}}$$

where $p_i$ means GCM simulation, $\overline{p}$ refers to the zonal average of that simulation, $q_i$ is the reference data, $\overline{q}$ refers to the zonal mean of reference data, and $n$ is the total time series.

**TC 8 Lines 1-2 (page 5): For clarity, rephrase the sentence "By comparing all analysis months.................was implemented" to "To implement the scoring scheme, a particular GCM's average of the Scorr and RMSE obtained by considering the analyses from all the months were compared to the mean values of the 'goodness-of-fit' metrics (Scorr and RMSE) of all the GCMs."**
**AC:** Changed.

**TC 9 Line 6 (page 5): what is "ceratain"?**
**AC:** modified to "certain".

**TC 10 Line 10 (page 5): "....we excluded GCMs that did not have a precipitation score of 1.....". It is possible that a particular GCM can have two or more simulation runs e.g. gfdl_cm2_1 and gfdl_cm2_0. I find it confusing whether the authors excluded the whole of such GCM or specifically the simulation runs with unsatisfactory scores. In case the entire GCM but not the simulation runs were excluded, which GCMs were those discarded? If the main purpose of the proposed method was bias correction, why should the GCMs with large bias be discarded? How can the authors verify the efficacy of their proposed method if they ideally want to use the GCMs with minimum bias? The answers to these questions should be presented clearly.**
**AC:** All of GCMs and its simulation runs were considered separately in the proposed method. This meant gfdl_cm2_1 and gfdl_cm2_0 were considered as two simulation runs of GCM and any GCM runs were discarded if they did not have a precipitation score of 1. The GCMs with large bias should be discarded because if the GCMs with low total scores meant the GCMs simulated parameters including precipitation showed the awkward or unmanageable output over the target basin. For the convincing future trend

of change in the basin-level assessment, the poor GCMs which total scores of lower than 4 or 5 (out of 7 or 8 parameters) are neglected.

**TC 11 Lines 18 (page 5), 33 (page 2), 1 and 20 (page 3), 6 (page 4), 1 (page 1) ....etc, : "We did this and that...." such colloquial words do not have spaces for their accommodation in papers to be published by a top journal like HESS.**
**AC:** All sentences are changed to passive form**.**
Line 18 (page 5) "A three-step statistical bias correction was proposed to eliminate these major GCM flaws."
Line 33 (page 2) In this way, a comprehensive and integrated statistical bias correction and spatial downscaling method together with tackling poor GCMs concern was developed.
Line 1 (page 3) Climate data from the Intergovernmental Panel on Climate Change (IPCC) Fourth Assessment Report (AR4) Coupled Model Inter-comparison Project 3 (CMIP3) under Special Report on Emission Scenarios 1 (SRES1) was used in this study.
Line 20 (page 3) Daily precipitation from 21-station were provided from the Philippines Atmospheric, Geophysical and Astronomical Services Administration and Metropolitan Waterworks and Sewerage System.
Line 6 (page 4) Thus, a unique benchmark and a scoring system for removal of poor GCMs is inevitable.
Line 1 (page 1) A three-step statistical bias correction method is introduced to solve global climate model (GCM) bias after excluding the improper GCMs through the decisive scoring scheme.

**TC 12 Line 23 (page 5): replace "it just considers the top as extremes in a year basic, in other words" with "by considering the maximum event in each year, the"**
**AC:** Replaced.

**TC 13 Line 24 (page 5): replace "year?s" with " year's"**
**AC:** Replaced.

**TC 14 Line 25-26 (page 5): ".......20 maxima (1981-2000) by defining extremes that were larger than the smallest one." This sentence is unclear. Do you mean the Annual Maxima Series (AMS) extracted from 20-year (1981-2000) data? "... larger than the smallest one." which smallest one? Still if you refer to AMS, it is well-known that for the AMS extraction, the largest event in each hydro-meteorological year is selected.**

**AC:** Changed from "the smallest one" to "the smallest one of 20 maxima". All of rain intensities larger than the minimum of 20 maxima were defined as extremes in AMS analysis.

**TC 15 Line 27 (page 5): in using Partial Duration Series (PDS), the authors should clearly present the criteria they used to ensure that the key requirement of frequency analysis (viz the extreme events to be independent and identically distributed) was fulfilled.**
**AC:** Added "Moreover, samples are independent each other as well as in their time of occurrence (Hershfield,1973)."

**TC 16 Line 28 (page 5): replace "of maxima" with " the extreme events"**
**AC:** Replaced.

**TC 17 Line 28 (page 5): change "occurred" to " occur"**
**AC:** Changed.

**TC 18 Line 29 (page 5): insert "quantile" between "improve" and "estimation"**
**AC:** Inserted.

**TC 19 Line 30 (page 5): change "rain" to "rainfall intensity"**
**AC:** Changed in line 33 on page 5.

**TC 20 Lines 1-3 (page 6): delete the sentences "This is because all rain..........................bias correction efficiency". Replace "Therefore, the" with "The"**
**AC:** Deleted and replaced.

**TC 21 Line 4 (page 6): replace "desirable" with "applied" and put a full stop after "correction" not a comma**
**AC:** Replaced.

**TC 22 Line 4 (page 6): replace "and" with "The GPD was used because"**
**AC:** Replaced.

**TC 23 Line 5 (page 6): delete "variables using"**

**AC:** Deleted.

**TC 24 Line 5 (page 6): replace "annual maximum flood" with "peak high flows"**
**AC:** Replaced.

**TC 25 Equation 3 (page 6): define the symbol $\xi$ as it appears the first time (not as you did later in line 7 of page 7). What about x? is it similar to that used and defined in equations 1 and 2? The authors should make this clear.**
**AC:** Added $\xi$ definition as location parameter in line 20 on page 6. x and y from equation 1 and 2 are changed to p and q.

$$S_{corr} = \frac{\sum_{i=1}^{n}\left(p_i - \bar{p}\right)\left(q_i - \bar{q}\right)}{\sqrt{\sum_{i=1}^{n}\left(p_i - \bar{p}\right)^2 \left(q_i - \bar{q}\right)^2}}$$

$$RMSE = \sqrt{\frac{1}{n}\sum_{i=1}^{n}\left(p_i - q_i\right)^2}$$

**TC 26 Line 17 (page 7): replace "tuned for the best fit" with "taken to be indicative of the best performance by the GCM".**
**AC:** Replaced.

**TC 27 Line 24 (page 7): replace "We solved this problem" with "Attempt to apply bias correction for the frequency of wet days was made by".**
**AC:** Replaced.

**TC 28 Line 26 (page 7): change "rain days" to "wet days". Implement this correction throughout the manuscript.**
**AC:** Changed "rain days" to "wet days" through the manuscript.

**TC 29 Line 27 (page 7): Is the word "beyond" the same as "above"? If so, change it accordingly.**
**AC:** Modified as "If GCM rainfall is smaller than that threshold, it is changed to zero for correction."

**TC 30 Line 22 (page 8): replace "just get rid of" with "minimize"**
**AC:** Replaced.

**TC 31 Line 9 (page 8): replace perfect"" with "reasonable"**
AC: Line3 on page 9 "perfect" is replaced with "reasonable".

**TC 32 Line 29 (page 9): what is "?a?" ?**
AC: Modified to ""a"".

**TC 33 Line 25 (page 11): what is basin?level?**
AC: Changed to basin-level.

**TC 34 Figure 5 (page 22): these plots are neither so informative nor scientifically convincing with respect to the extreme value analysis which the authors claim to be considering. Since comparison of the quantiles from extremes extracted based on the AMS and PDS are being compared, why can't the quantile plots be made instead of showing the number of days and rainfall intensity? I recommend better plots than those in Figure 5 be made e.g. extreme rainfall intensity versus log-transformed return periods for better quantile-based assessment of bias. For an example of such plots, the authors can see Figure 3 of Sunyer et al. (2012), Figure 2 of Willems and Vrac (2011), Figure 4 of Onyutha et al. (2016), etc.**
AC: Figure 5 (a) and (b) (Fig.2 in the interactive comment) compares the frequency of extreme events from two series of gfdl_cm2-0 GCM. One series is defined by all of rain intensities larger than the smallest intensity of 20 annual maximum series (AMS) in Fig. 5a and the other one is based on PDS over the predefined threshold of GPD as in Fig. 5b.

By using adjusted annual maximum series (AMS) to suitable distribution such as Gumble, log-normal or Weibull, bias correction of extreme is carried out and quantiles are estimated. Other PDS series with GPD correction is also done and quantiles estimates are different in two series but both of them are comparable to observation. But the big difference between the two series is the number of extreme days according to each definition and not the estimated quantiles. This error contributes the large variation in the mean of bias-corrected long-term monthly intensity during 1981-2000 after three-steps bias correction. The reason is an inclusive of the excessive number of extremes in bias-corrected GCM rain series by AMS and an unequal distribution of extremes in each climatology month because statistical bias correction has no time references. Therefore, the extreme events defined by AMS series is not used in bias correction.

According to the reason mentioned above, Figure 5 is modified to exceedance probability comparison between the two series before bias correction and after bias

correction compared to observed data at CLSU station.

**TC 35 Line 29 (page 13): replace "eliminates" with "reduces"**
**AC:** Replaced.

**TC 36 The maps in Figures 1, and 16-19 should be presented with clearly marked grids and graticules to show locations (degrees of latitude and longitude) in geographic coordinates.**
**AC:** Modified Fig.1 (Fig.3 in the interactive comment) and 16-19 with decimal degree grids and 16-19 are combined as Fig. 17 (Fig.4 in the interactive comment).

**TC 37 For Figures 9 and 10 it cannot be understood that the letters a, b, ...and g in the horizontal axis represent the IDs of the GCMs as presented in Table 3 though stated in lines 28-29 of page 9. Make this clear in the Figure caption as well. No any difference seems noticeable by comparing the plot for the GCM 'a' before and after the application of the bias correction. How can the authors explain this realization? This should be clarified within the text of second paragraph in section of 3.4.**
**AC:** Modify in line14 on page 10 as follows: "In both figures, label "a" on the x-axis is for observation and "b, c, d, e, f, g" on that axis refers to the selected 6 GCMs from Table 3. GCMs lists under Yoshio (the third column) are for Matsuyama and under Medjerda (the last column) is for Oued Mellegue."

Added in Figure 9 caption as follows:
  "Label "a" on the x-axis is for observation and "b, c, d, e, f, g" on that axis refers to the selected 6 GCMs under Yoshino (the third column) in Table 3."

Added in Figure 10 caption as follow:
"Label "a" on the x-axis is for observation and "b, c, d, e, f, g" on that axis refers to the selected 6 GCMs under Medjerda (the last column) in Table 3."

**TC 38 Figure 14: compute the exceedance probability of each extreme rainfall event and use it to replace the ranking order plotted on the horizontal axis. I also recommend that throughout the manuscript, the expression "ranking order statistics" be replaced with "exceedance probability".**
**AC:** Modified X-axis with "Exceedance probability" in Figure 14. Replace "ranking

order statistics" to "exceedance probability" throughout the manuscript.

**References**

Argüeso, D., Evans, J., and Fita, L.: Precipitation bias correction of very high resolution regional climate models, Hydrology and Earth System Sciences Discussions, 10, 8145–8165, 2013.

Bordoy, R. and Burlando, P.: Bias correction of regional climate model simulations in a region of complex orography, Journal of Applied Meteorology and Climatology, 52, 82–101, 2013.

de Zea Bermudez, P., and Turkman, M. A.: Bayesian approach to parameter estimation of the generalized Pareto distribution. Test, 12(1), 259-277, 2003.

Graham, L. P., Andréasson, J., and Carlsson, B.: Assessing climate change impacts on hydrology from an ensemble of regional climate models, model scales and linking methods–a case study on the Lule River basin, Climatic Change, 81, 293–307, 2007.

Hay, L. E., Wilby, R. L., and Leavesley, G. H.: A comparison of delta change and downscaled GCM scenarios for three mounfainous basins in THE UNITED STATES1, 2000.

Hosking, J. R., and Wallis, J. R.: Parameter and quantile estimation for the generalized Pareto distribution. Technometrics, 29(3), 339-349, 1987.

Hosking, J. R., and Wallis, J. R.: Parameter and quantile estimation for the generalized Pareto distribution. Technometrics, 29(3), 339-349, 1987.

Hurkmans, R., Terink, W., Uijlenhoet, R., Torfs, P., Jacob, D., and Troch, P. A.: Changes in streamflow dynamics in the Rhine basin under three high-resolution regional climate scenarios, Journal of Climate, 23, 679–699, 2010.

Ines, A. V. and Hansen, J. W.: Bias correction of daily GCM rainfall for crop simulation studies, Agricultural and forest meteorology, 138, 35 44–53, 2006.

Kawase, H., Yoshikane, T., Hara, M., Kimura, F., Yasunari, T., Ailikun, B., ... & Inoue, T. (2009). Intermodel variability of future changes in the Baiu rainband estimated by the pseudo global warming downscaling method. Journal of Geophysical Research: Atmospheres, 114(D24).

Lafon, T., Dadson, S., Buys, G., and Prudhomme, C.: Bias correction of daily precipitation simulated by a regional climate model: a comparison of methods, International Journal of Climatology, 33, 1367–1381, 2013.

Leander, R. and Buishand, T. A.: Resampling of regional climate model output for the simulation of extreme river flows, Journal of Hydrology, 332, 487–496, 2007.

Li, H., Sheffield, J., and Wood, E. F.: Bias correction of monthly precipitation and temperature fields from Intergovernmental Panel on Climate Change AR4 models using equidistant quantile matching, Journal of Geophysical Research: Atmospheres (1984–2012), 115, 2010.

Madsen, H., Pearson, C. P., and Rosbjerg, D.: Comparison of annual maximum series and partial duration series methods for modeling extreme hydrologic events: 2. Regional modeling. Water

resources research, 33(4), 759-769, 1997.

Martins, E. S., and Stedinger, J. R.: Generalized maximum likelihood Pareto-Poisson estimators for partial duration series. Water Resources Research, 37 (10), 2551-2557, 2001.

Maurer, E., Das, T., and Cayan, D.: Errors in climate model daily precipitation and temperature output: time invariance and implications for bias correction, Hydrology and Earth System Sciences Discussions, 10, 1657–1691, 2013.

Onyutha, C., Tabari, H., Rutkowska, A., Nyeko-Ogiramoi, P., and Willems, P.: Comparison of different statistical downscaling methods for climate change rainfall projections over the Lake Victoria basin considering CMIP3 and CMIP5, Journal of Hydro-environment Research, 12, 31–45, 2016.

Piani, C. and 5 Haerter, J.: Two dimensional bias correction of temperature and precipitation copulas in climate models, Geophysical Research Letters, 39, 2012.

Piani, C., Haerter, J., and Coppola, E.: Statistical bias correction for daily precipitation in regional climate models over Europe, Theoretical and Applied Climatology, 99, 187–192, 2010a.

Rasmy, M., Tanda, C., Jaranilla-Sanchez, P. A., Koike, T., Hara, M., Fujita, M., & Kawase, H. (2013). Identifying gaps and opportunities between statistical and dynamical downscaling approaches over Shikoku island, Japan. Annual Journal of JSCE Hydraulic Eng., 69(4), I_133-I_138.

Rosbjerg, D., Madsen, H., and Rasmussen, P. F.: Prediction in partial duration series with generalized pareto-distributed exceedances. Water Resources Research, 28(11), 3001-3010, 1992.

Schoetter, R., Hoffmann, P., Rechid, D., and Schlünzen, K. H.: Evaluation and bias correction of regional climate model results using model evaluation measures, Journal of Applied Meteorology and Climatology, 51, 1670–1684, 2012.

Sunyer Pinya, M. A., Hundecha, Y., Lawrence, D., Madsen, H., Willems, P., Martinkova, M., Vormoor, K., Bürger, G., Hanel, M., Kriaučiuniene, J., et al.: Inter-comparison of statistical downscaling methods for projection of extreme precipitation in Europe, Hydrology and Earth System Sciences, 19, 1827–1847, 2015.

Sunyer, M., Madsen, H., and Ang, P.: A comparison of different regional climate models and statistical downscaling methods for extreme rainfall estimation under climate change, Atmospheric Research, 103, 119–128, 2012.

Tajvidi, N.: Confidence intervals and accuracy estimation for heavy-tailed generalized Pareto distributions. Extremes, 6 (2), 111-123, 2003.

Teutschbein, C. and Seibert, J.: Is bias correction of regional climate model (RCM) simulations possible for non-stationary conditions? Hydrology and Earth System Sciences, 17, 5061–5077, 2013.

Wood, A.W., Leung, L. R., Sridhar, V., and Lettenmaier, D.: Hydrologic implications of dynamical and statistical approaches to downscaling climate model outputs, Climatic change, 62, 189–216, 2004.